# Differential methylation analysis in neuropathologically confirmed dementia with Lewy bodies

Paolo Reho[1,2], Sara Saez-Atienzar [3], Paola Ruffo[3,4], Sultana Solaiman[1], Zalak Shah[1], Ruth Chia[3], Karri Kaivola [1], Bryan J. Traynor [3], Bension S. Tilley[5], Steve M. Gentleman [5], Angela K. Hodges [6], Dag Aarsland [6,7], Edwin S. Monuki[8], Kathy L. Newell[9], Randy Woltjer[10], Marilyn S. Albert[11], Ted M. Dawson[11,12,13,14], Liana S. Rosenthal[11], Juan C. Troncoso[15], Olga Pletnikova [16], Geidy E. Serrano[17], Thomas G. Beach[17], Hariharan P. Easwaran[18] & Sonja W. Scholz [1,11✉]

Dementia with Lewy bodies (DLB) is a common form of dementia in the elderly population. We performed genome-wide DNA methylation mapping of cerebellar tissue from pathologically confirmed DLB cases and controls to study the epigenetic profile of this understudied disease. After quality control filtering, 728,197 CpG-sites in 278 cases and 172 controls were available for the analysis. We undertook an epigenome-wide association study, which found a differential methylation signature in DLB cases. Our analysis identified seven differentially methylated probes and three regions associated with DLB. The most significant CpGs were located in *ARSB* (cg16086807), *LINC00173* (cg18800161), and *MGRN1* (cg16250093). Functional enrichment evaluations found widespread epigenetic dysregulation in genes associated with neuron-to-neuron synapse, postsynaptic specialization, postsynaptic density, and CTCF-mediated synaptic plasticity. In conclusion, our study highlights the potential importance of epigenetic alterations in the pathogenesis of DLB and provides insights into the modified genes, regions and pathways that may guide therapeutic developments.

[1] Neurodegenerative Diseases Research Unit, National Institute of Neurological Disorders and Stroke, Bethesda, MD, USA. [2] Laboratory of Precision Environmental Health, Department of Environmental Health Sciences, Mailman School of Public Health, Columbia University, New York, NY, USA. [3] Neuromuscular Diseases Research Section, National Institute on Aging, Bethesda, MD, USA. [4] Medical Genetics Laboratory, Department of Pharmacy, Health and Nutritional Sciences, University of Calabria, Rende, Italy. [5] Neuropathology Unit, Department of Brain Sciences, Imperial College London, London, UK. [6] Institute of Psychiatry, Psychology and Neuroscience, King's College London, London, UK. [7] Centre for Age-Related Medicine, Stavanger University Hospital, Stavanger, Norway. [8] Department of Pathology & Laboratory Medicine, School of Medicine, University of California Irvine, Irvine, CA, USA. [9] Department of Pathology and Laboratory Medicine, Indiana University School of Medicine, Indianapolis, IN, USA. [10] Department of Neurology, Oregon Health & Sciences University, Portland, OR, USA. [11] Department of Neurology, Johns Hopkins University Medical Center, Baltimore, MD, USA. [12] Neuroregeneration and Stem Cell Programs, Institute of Cell Engineering, Johns Hopkins University School of Medicine, Baltimore, MD, USA. [13] Department of Pharmacology and Molecular Sciences, Johns Hopkins University School of Medicine, Baltimore, MD, USA. [14] Solomon H. Snyder Department of Neuroscience, Johns Hopkins University School of Medicine, Baltimore, MD, USA. [15] Department of Pathology (Neuropathology), Johns Hopkins University Medical Center, Baltimore, MD, USA. [16] Department of Pathology and Anatomical Sciences, Jacobs School of Medicine and Biomedical Sciences, University at Buffalo, Buffalo, NY, USA. [17] Civin Laboratory for Neuropathology, Banner Sun Health Research Institute, Sun City, AZ, USA. [18] The Sidney Kimmel Comprehensive Cancer Center at Johns Hopkins, Baltimore, MD, USA. ✉email: sonja.scholz@nih.gov

Dementia with Lewy bodies (DLB) is a heterogeneous neurodegenerative disease characterized by parkinsonism, visual hallucinations, fluctuating mental status, and REM-sleep behavior disorder[1]. There are an estimated 1.4 million cases living in the United States[2], and current therapy is limited to symptomatic and supportive care. Although genetic research studies have identified heritable factors that are important in the etiology of this understudied disease[3,4], the molecular causes remain poorly understood, and little is known about non-genetic contributors to its pathogenesis.

Epigenetic changes are modifications to the DNA that regulate gene expression. These modifications are influenced by aging, the environment, lifestyle, disease state, and other factors. They allow cells to respond dynamically to the outside world and are considered the interface between genetic and environmental components. One type of epigenetic alteration is DNA methylation. This tissue-specific mechanism occurs when a methyl group is transferred onto a cytosine, mostly occurring in the context of a cytosine-phosphate-guanine dinucleotide sequence (CpG) in higher eukaryotes. These modifications change the DNA accessibility to the transcriptional machinery complex and fine-modulate gene expression. Importantly, epigenetic modifications are thought to play a prominent role in age-associated neurological diseases, such as Alzheimer's disease and Parkinson's disease[5–7]. Evidence is also emerging that DNA methylation changes influence the risk of developing DLB[4,8–10].

In this study, we investigated the role of DNA methylation in the pathogenesis of DLB. We performed an epigenome-wide association study (EWAS) to characterize the differential methylation patterns in cerebellar tissue obtained from 298 pathologically confirmed DLB cases and 203 neurologically healthy controls. We further performed gene-region and pathway analyses, demonstrating widespread epigenetic changes associated with this fatal neurodegenerative disease.

## Results

**Epigenome-wide association study design**. We performed an EWAS using cerebellar brain tissue obtained from patients diagnosed with DLB and healthy individuals. We profiled their DNA methylation status with the Illumina MethylationEPIC arrays. After quality control filters were applied, 728,197 sites were tested for association with DLB in 278 cases and 172 controls. We then designed the regression model adjusting the data by age, sex, experimental batch, five principal components from genotyping data, cell type proportion, and 44 surrogate variables from methylation data.

**EWAS identifies differentially methylated probes**. We identified seven differentially methylated probes (DMPs) that surpassed the genome-wide significance threshold (Bonferroni adjusted $p$ value < 0.05) (Fig. 1, Table 1). Among these, four probes were hypomethylated (cg16086807, cg04866173, cg11099930, and cg24435966) and three were hypermethylated (cg18800161, cg16250093, and cg06951630) in DLB patients compared to healthy controls (Fig. 2). Overall, we detected mild epigenetic modulations, where the differences between β-values (Δβ) in cases and controls ranged from −0.10 to 0.091 (Fig. 1). Six out of the seven DMPs were mapped within gene regions, overlapping the gene body of $ARSB$ (cg16086807, Bonferroni-adjusted $p$ value = 1.19E-03, Δβ = −0.020), $LINC00173$ (cg18800161, adjusted $p$ value = 5.26E-03, Δβ = 0.006), $MGRN1$ (cg16250093, adjusted $p$ value = 1.02E-02, Δβ = 0.008), $FHL2$ (cg04866173, adjusted $p$ value = 1.15E-02, Δβ = −0.023), $IQSEC1$ (cg06951630, adjusted $p$ value = 3.62E-02, Δβ = 0.009), and the promoter region of $NIPBL$ (cg11099930, adjusted $p$ value = 1.50E-02, Δβ = −0.010) (Table 1).

Among the DMP-associated genes, three have been previously implicated in neurodegenerative or neurodevelopmental disorders: $ARSB$, encoding for the Arylsulfatase B, has been associated with Alzheimer's disease and Parkinson's disease;[11,12] $MGRN1$, encoding Mahogunin Ring Finger 1, has been implicated in late-onset spongiform neurodegeneration;[13] and $IQSEC1$, encoding for the IQ Motif And Sec7 Domain ArfGEF 1, has been associated with a neurodevelopmental disorder[14].

Our data showed modest inflation (lambda = 1.19). To further explore inflation, we also corrected the $p$ values using the Bioconductor package $bacon$ (lambda = 1.02). This approach replicated the results of the main study, where all seven DMPs reached the FDR threshold and the top two probes surpassed the genome-wide significance threshold based on Bonferroni correction (Supplementary Fig. 1).

**Genes related to neurodegenerative diseases identified among the sub-significant signals**. In addition to the top seven DMPs, our analysis identified 41 probes surpassing the less conservative false discovery rate (FDR) threshold of 0.05 (Supplementary Data 1). These sub-significant probes were localized within genes previously associated with neurological disorders: $EPS8$ (cg06658698, FDR adjusted $p$ value = 1.49E-02, Δβ = 0.030), $ARL6IP1$ (cg20872370, FDR adjusted $p$ value = 2.24E-02, Δβ = −0.007), $FRMD4A$ (cg03775372, FDR adjusted $p$ value = 2.82E-02, Δβ = −0.020), and $PIAS1$ (cg05751215, FDR adjusted $p$ value = 3.40E-02, Δβ = −0.009). In particular, $EPS8$, encoding for the Epidermal Growth Factor Receptor Pathway Substrate 8, and $FRMD4A$, that encodes for FERM Domain Containing 4 A, have been recently associated with Alzheimer's disease; [15,16] $ARL6IP1$, encoding for ADP Ribosylation Factor Like GTPase 6 Interacting Protein 1, has been associated with hereditary spastic paraplegia;[17] and $PIAS1$, that encodes for Protein Inhibitor Of Activated STAT 1, has been implicated in Huntington's disease[18].

**Identification of differentially methylated genomic regions**. Recent research has shown that differentially methylated regions (DMRs) are more highly associated with diseases than differential methylation at individual CpG sites alone[19]. For this reason, we examined DMRs in our case-control dataset. Our analysis identified 32 CpG sites that clustered in three different DMRs characterized by a hypermethylation signature in DLB cases compared to controls (Table 2). For example, we identified a DMR overlapping the promoter region of $DHRS4$ and its antisense lncRNA $DHRS4-AS1$ (adjusted $p$ value = 1.59E-10, mean $\Delta\beta$ = 0.022). $DHRS4$ has been recently suggested as a novel risk gene for inducing neurodegeneration in mouse models of amyotrophic lateral sclerosis[20].

**Sex-specific EWAS delineates a male-driven effect**. To explore differential methylation patterns among men and women, we performed an interaction model comparing male study participants ($n = 165$ DLB cases and $n = 122$ controls) and female subjects ($n = 113$ DLB cases and $n = 50$ controls). We identified five differentially methylated probes showing a statistically significant epigenetic modulation in males compared to females (Bonferroni adjusted $p$ value < 0.05) (Fig. 3, Supplementary Data 2). All five DMPs were also identified in the overall cohort analysis. The most associated site in males was located 461 base pairs upstream of $RP11-1E6.1$ (cg24435966, Bonferroni-adjusted $p$ value = 4.71E-04, Δβ = −0.037), a predicted lncRNA with unknown function. The four male-specific DMPs overlapped the following genes: $ARSB$ (cg16086807, adjusted $p$ value = 7.39E-04, Δβ = −0.022), $FRMD4A$ (cg03775372, adjusted $p$ value = 1.17E-02, Δβ = −0.026), $LINC00173$ (cg18800161, adjusted $p$ value = 1.77E-02, Δβ = 0.007),

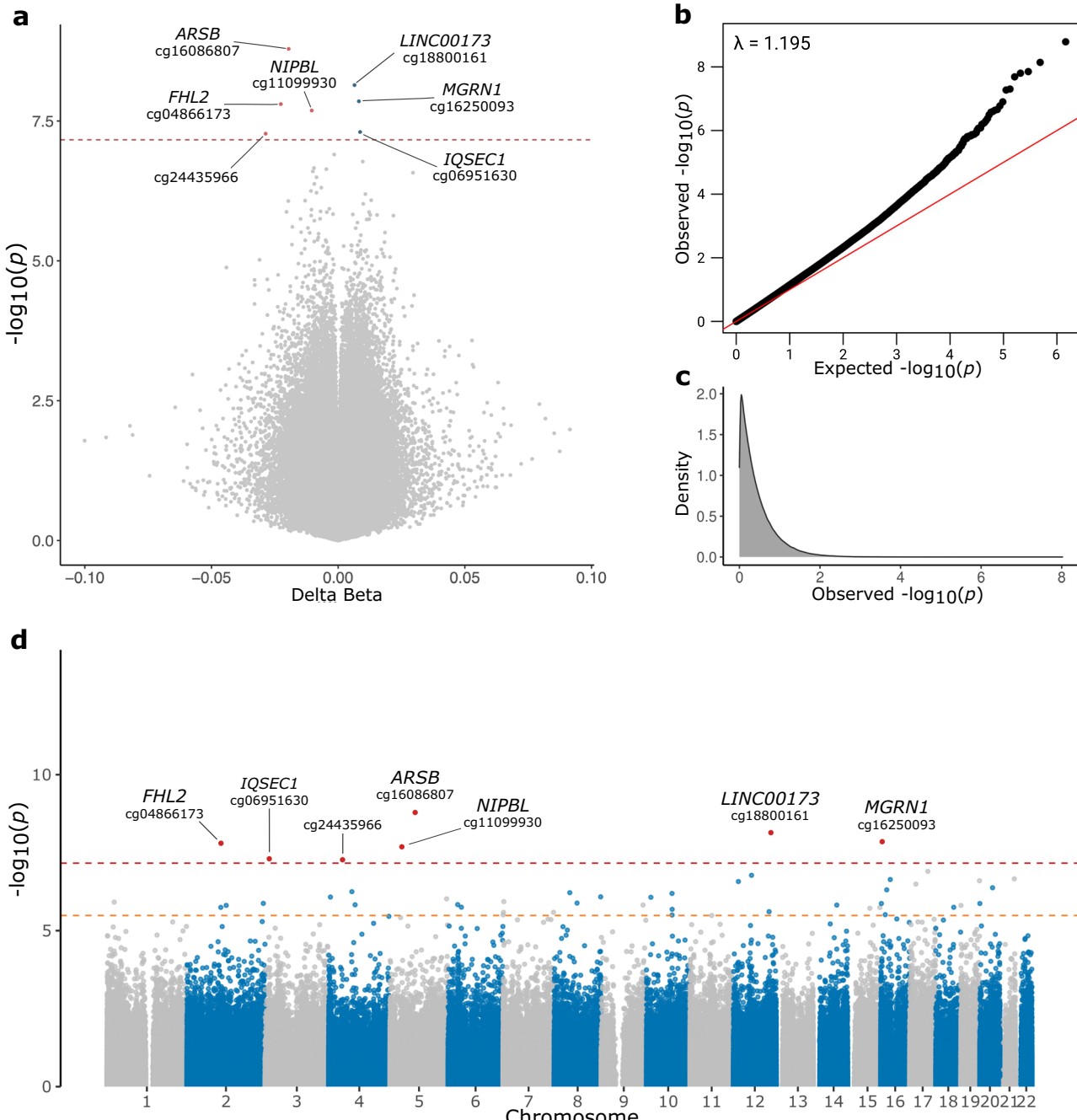

**Fig. 1 Volcano plot, QQ-plot, and Manhattan plot of EWAS results.** Volcano plot (**a**) showing statistical significance (-log₁₀ *p* value) and magnitude of change (delta beta, *Δβ*) of all CpG sites included in the DLB EWAS analysis. Red and blue dots indicate significantly hypomethylated DMPs and hypermethylated sites in DLB cases compared to controls, respectively. The Bonferroni adjusted *p* value < 0.05 threshold is shown as a red dashed line. QQ-plot (**b**) showing the *p* value distribution and inflation (lambda value). Density plot (**c**) illustrating the observed *p* value distribution. Manhattan plot (**d**) showing the *p* values of the tested probes across the genome. The genome-wide significance threshold (Bonferroni adjusted *p* value < 0.05) is shown as a red dashed line, while the orange dashed line represents the FDR threshold. Probes surpassing the genome-wide significance are shown as red dots.

and *NIPBL* (cg11099930, adjusted *p* value = 4.88E-02, Δβ = −0.011). Finally, our sex-specific EWAS did not identify any significant differentially methylated regions.

**Functional enrichment analysis determined pathways associated with DLB.** We performed a pathway enrichment analysis of differentially methylated genes in our case-control cohort. Specifically, we investigated the 43 genes that overlapped FDR-significant differentially methylated probes. This analysis detected eight biological processes, cellular components, and molecular functions that

were significantly associated with DLB. Among the Gene Ontology (GO) terms, we identified an association with the terms "regulation of response to stimulus" (Bonferroni-adjusted *p* value = 0.0237), "neuron to neuron synapse" (adjusted *p* value = 0.0494), "vesicle" (adjusted *p* value = 0.0489), "postsynaptic specialization" (adjusted *p* value = 0.0385), and "postsynaptic density" (adjusted *p* value = 0.0276) (Fig. 4). Of note, 29 out of 44 genes showed an interaction with the CTCF protein, a transcription factor that has been associated with Alzheimer's disease and synaptic organization (adjusted *p* value = 0.0011) (Table 3)[21,22].

**Table 1 Significant differentially methylated probes in the DLB EWAS.**

| Probe ID | Chr | Position | Gene | Gene Region | Δβ | P value | Adj.P | Δβ Males vs Females | P value Males vs Females | Adj.P.Males vs Females |
|---|---|---|---|---|---|---|---|---|---|---|
| cg16086807 | 5 | 78,271,020 | ARSB | Body | −0.020 | 1.63E-09 | 1.19E-03 | −0.037 | 1.02E-09 | 7.39E-04 |
| cg18800161 | 12 | 116,971,933 | LINC00173 | Body | 0.006 | 7.22E-09 | 5.26E-03 | 0.007 | 2.432E-08 | 1.77E-02 |
| cg16250093 | 16 | 4,691,998 | MGRN1 | Body | 0.008 | 1.41E-08 | 1.02E-02 | 0.008 | 2.34E-07 | 1.71E-01 |
| cg04866173 | 2 | 105,990,524 | FHL2 | Body | −0.023 | 1.58E-08 | 1.15E-02 | −0.023 | 2.98E-07 | 2.17E-01 |
| cg11099930 | 5 | 36,876,680 | NIPBL | TSS200 | −0.010 | 2.06E-08 | 1.50E-02 | −0.011 | 6.71E-08 | 4.88E-02 |
| cg06951630 | 3 | 13,249,045 | IQSEC1 | Body | 0.009 | 4.97E-08 | 3.62E-02 | 0.009 | 3.94E-07 | 2.87E-01 |
| cg24435966 | 4 | 43,342,431 | – | – | −0.029 | 5.33E-08 | 3.89E-02 | −0.037 | 6.47E-10 | 4.71E-04 |

*Chr.* chromosome, *Adj.P* adjusted *p* value
Differentially methylated probes in the DLB EWAS. Chromosome positions are shown relative to the human reference genome (hg19). Gene names are shown according to UCSC RefGen. The *Δβ* values refer to the difference between DNA methylation (*β*-values) in cases compared to controls (e.g., −0.020 indicates that DLB cases show a 2% decrease in DNA methylation compared to controls). Adjusted *p* value refer to Bonferroni corrected *p* values.

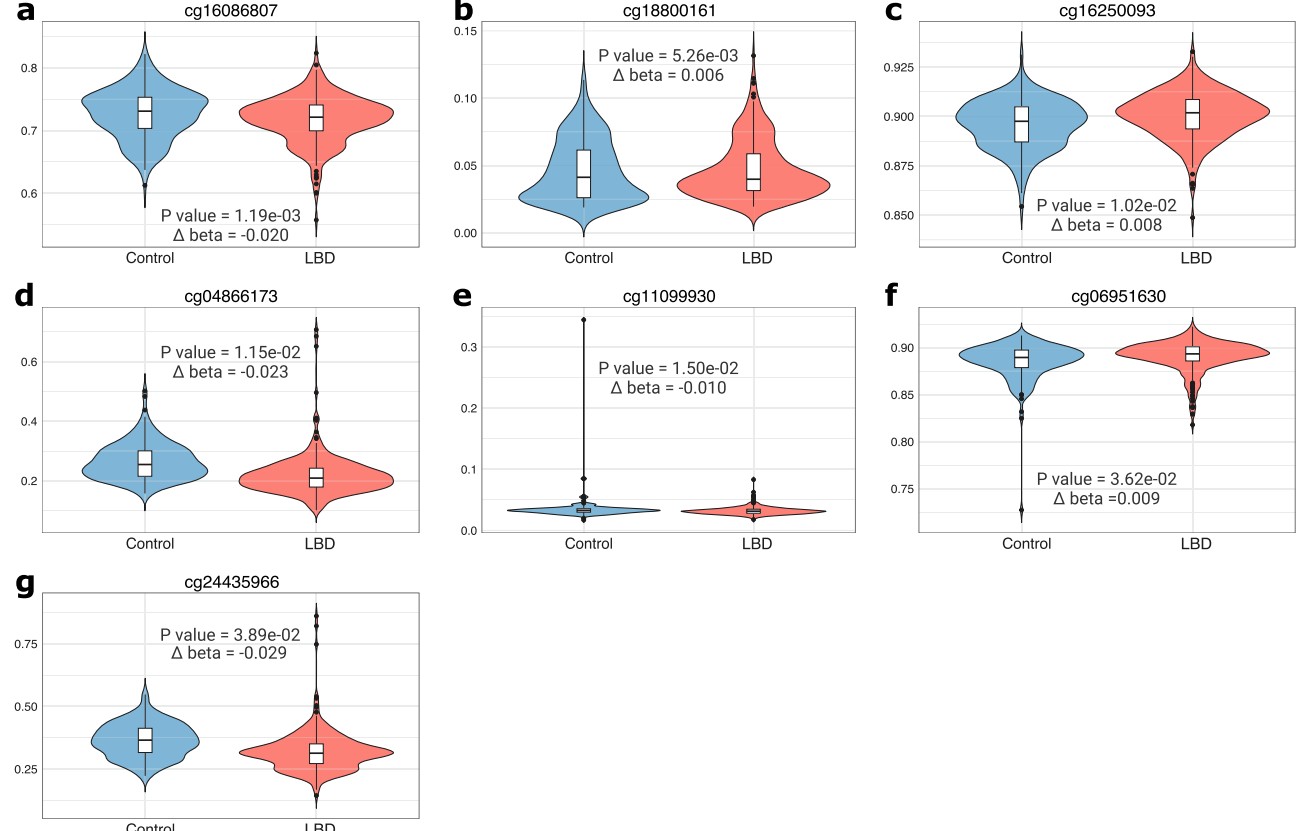

**Fig. 2 Differentially methylated probes in DLB.** The violin plots show the DNA methylation (beta value) distribution (violin shape) in the seven differentially methylated probes (panels **a–g**). The vertical axis represents the range of values in the dataset, where 0 and 1 mean fully unmethylated and fully methylated respectively. The box plot represents the interquartile range of the dataset (25% bottom, 75% top), the middle line represents the median of the distribution, and the central line shows the value distribution. Black dots represent outliers. The overall difference between DLB cases and controls is shown as delta beta (*Δβ*); negative and positive values refer to hypomethylation and hypermethylation in DLB cases, respectively (e.g., −0.020 indicates that DLB cases show a 2% decrease in DNA methylation compared to controls). *P* values refer to Bonferroni corrected *p* values.

## Discussion

Our analyses illustrate the value of EWAS in unraveling the multiplex architectures of neurodegenerative conditions and highlight the potential contributions of methylation to the pathogenesis of DLB. We identified several probes that were differentially methylated in the DLB cases (Fig. 1). Interestingly, many of these genes are highly expressed in the brain (Fig. S2) and have been previously implicated in neurological disorders or central nervous system development.

Chief among the loci identified by our study was *ARSB*, encoding for Arylsulfatase B, a member of the sulfatase family that removes sulfate groups from chondroitin-4-sulfate, triggering its degradation. This lysosomal enzyme is involved in cell adhesion, migration, and invasion in colonic epithelium[23]. In cultured astrocytes, *ARSB* silencing increases chondroitin-4-sulfate and neurocan levels, and inhibits astrocyte-mediated neurite outgrowth, suggesting that *ARSB* may play an important role in neuronal plasticity in the central nervous system[24]. Homozygous or compound heterozygous mutations of the gene lead to Mucopolysaccharidosis type VI, a lysosomal storage disorder characterized by skeletal anomalies, short stature, and cardiac abnormalities[25]. Our data complement previous studies

**Table 2 Differentially methylated regions in the DLB EWAS.**

| Chr. | Start | End | Width (bp) | No. CpGs | P value | Mean Δβ | Overlapping Genes | Gene Function |
|---|---|---|---|---|---|---|---|---|
| 6 | 29,648,161 | 29,649,024 | 864 | 20 | 5.85E-11 | 0.048 | ZFP57 | Regulation of gene expression controlling DNA methylation |
| 14 | 24,422,749 | 24,422,956 | 208 | 6 | 1.59E-10 | 0.022 | DHRS4, DHRS4-AS1 | Cell metabolism |
| 1 | 183,154,778 | 183,155,154 | 377 | 6 | 4.37E-10 | 0.018 | LAMC2 | Cell adhesion, motor neuron axon guidance, epidermis development |

Chr. chromosome, No.CpGs number of CpGs per region, Mean Δβ mean probe delta beta.
Chromosome positions are shown relative to the human reference genome (hg19). P value refers to smoothed FDR corrected p value.

implicating *ARSB* variants in Alzheimer's disease and Parkinson's disease[11,12], expanding the pathogenic role of the gene in neurodegeneration.

DLB affects males more than females[26], but little is known about sex-specific contributions to the pathogenesis. Our sex-specific EWAS showed a male-driven effect, suggesting that epigenetic modifications could modulate the risk in males and females separately. However, our analysis did not include sex chromosomes, and future studies exploring epigenetic changes in these chromosomes are needed to draw conclusions.

Interestingly, our study also showed an enrichment of genes that regulate the neuron-to-neuron synapse, postsynaptic specialization, and postsynaptic density in DLB. Moreover, we highlighted the transcriptional repressor CTCF as a key factor able to orchestrate these biological processes. This protein is actively involved in maintaining the three-dimensional structure of the chromatin, creating topologically associated functional domains within the nucleus[27]. Our data suggest that the chromatin architecture could play a role in the pathogenesis of DLB, and support recent evidence implicating CTCF-mediated synaptic plasticity in Alzheimer's disease[22]. These observations expand the role of CTCF and synaptic organization-related genes in neurodegeneration.

Our data corroborate emerging evidence implicating aberrant epigenetic modulation in neurodegeneration[28,29]. Only a few studies have investigated the epigenetic changes associated with DLB, and they differ in sample size, targeted tissue, and study design[8,10,30]. For example, Shao and colleagues performed an EWAS of the Brodman area 7 of the brain in a cohort consisting of fifteen pathologically confirmed DLB cases and sixteen neurologically healthy controls. Despite the limited sample size and the different brain regions investigated, their study design does represent the closest structure to our EWAS. In contrast, Nasamran and collaborators profiled blood epigenetic modulations comparing 42 DLB patients and 50 Parkinson's disease dementia cases, while Pihlstrom and colleagues explored the epigenetic modulations associated with different Braak Lewy body disease stages in 322 Parkinson's disease and DLB cases. Our study identified none of the DMPs or DMRs previously associated with DLB (Supplementary Data 3). However, the outlined differences in the design of these studies may account for these discrepancies.

Significant changes in methylation have also been observed in Alzheimer's disease and Parkinson's disease[31–33]. Sharma and collaborators showed that epigenetic modifications across single nucleotide polymorphisms, located within the first intron of the *SNCA* gene, modulate the susceptibility to Parkinson's disease. A meta-analysis of 1,453 individuals with Alzheimer's disease, investigating the epigenetic changes associated with Braak neurofibrillary tangle stage, identified differentially methylated sites and regions in the prefrontal cortex, temporal gyrus, and entorhinal cortex, but not in the cerebellum. Interestingly, our EWAS replicated some of the findings from Smith's study, identifying epigenetic modification involving a common CpG site within *FRMD4A* (cg03775372) and a common DMP-associated gene, *MKL2*. This finding raises the scientific interest surrounding cg03775372 and the *FRMD4A* gene, since the same CpG reached the FDR significance in our main study and achieved genome-wide significance in the sex-specific EWAS.

We selected cerebellar tissue for our research as it is relatively spared in the terminal stages of DLB, unlike cortical tissue in which most cells of interest have been lost to neurodegeneration. Selecting a relatively spared tissue source provides a more accurate window into the epigenetic plasticity of a disease. This detail needs to be weighed against the fact that the disease-relevant changes are likely more prominent and representative in the

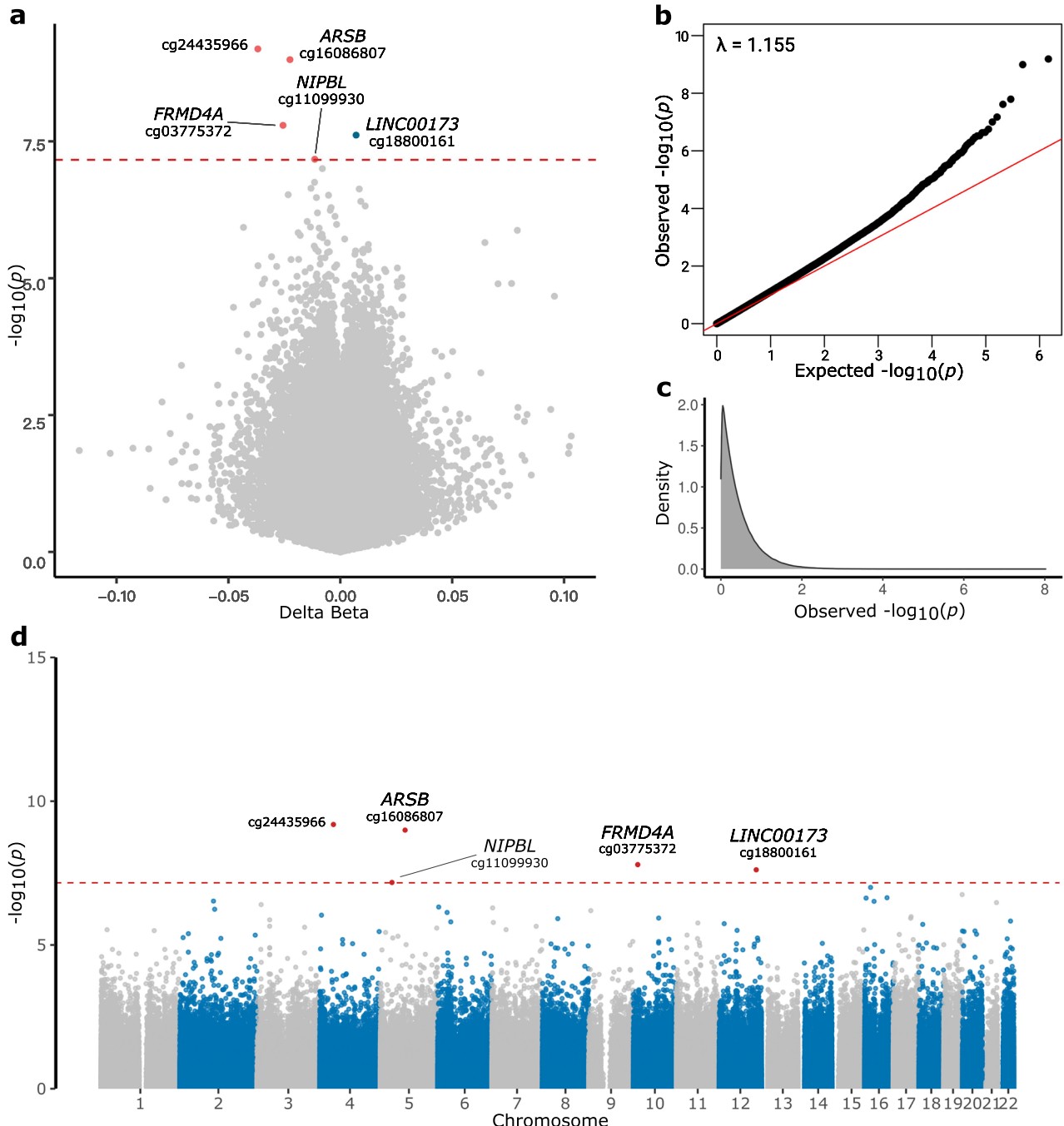

**Fig. 3 Volcano plot, QQ-plot, and Manhattan plot of sex-specific EWAS.** Volcano plot **a)** showing genome-wide significance (-log₁₀ p value) and magnitude of change (Δβ) of all sites included in the analysis. Negative values (dots on the left side of the volcano plot) indicate hypomethylated DMPs, and hypermethylated sites are displayed as positive values (right-sided dots). The Bonferroni adjusted p value < 0.05 threshold is shown as a red dashed line. QQ-plot (**b**) showing p values distribution and inflation (lambda value). Density plot (**c**) illustrating the observed p value distribution. Manhattan plot (**d**) showing the p value of the probes across the genome. The Bonferroni adjusted p value < 0.05 threshold is shown as a red dashed line. Probes surpassing the genome-wide significance are shown as red dots.

regions primarily affected by the disease. Sampling multiple regions for comparison would have been ideal. Future efforts will likely increase our ability to identify disease-associated methylation patterns across the brain, and we have made the EWAS results from our study publicly available to facilitate this unfolding research.

Our EWAS has several limitations. The Illumina MethylationEPIC array contains only a fraction of the human genome's CpGs. Furthermore, the bisulfite conversion chemistry does not distinguish

between 5-methylcytosine (5mC) and 5-hydroxymethylcytosine (5hmC), a brain-specific intermediate product of 5mC demethylation[34]. We also did not adjust the data for environmental factors that may impact the methylation status, such as vascular comorbidities, smoking, and alcohol use.

The *minfi*-approach that we used to estimate the cell type proportion was able to discriminate between neuronal and nonneuronal cells, using the frontal cortex region as reference. This method is based on the neuronal-specific protein NeuN,

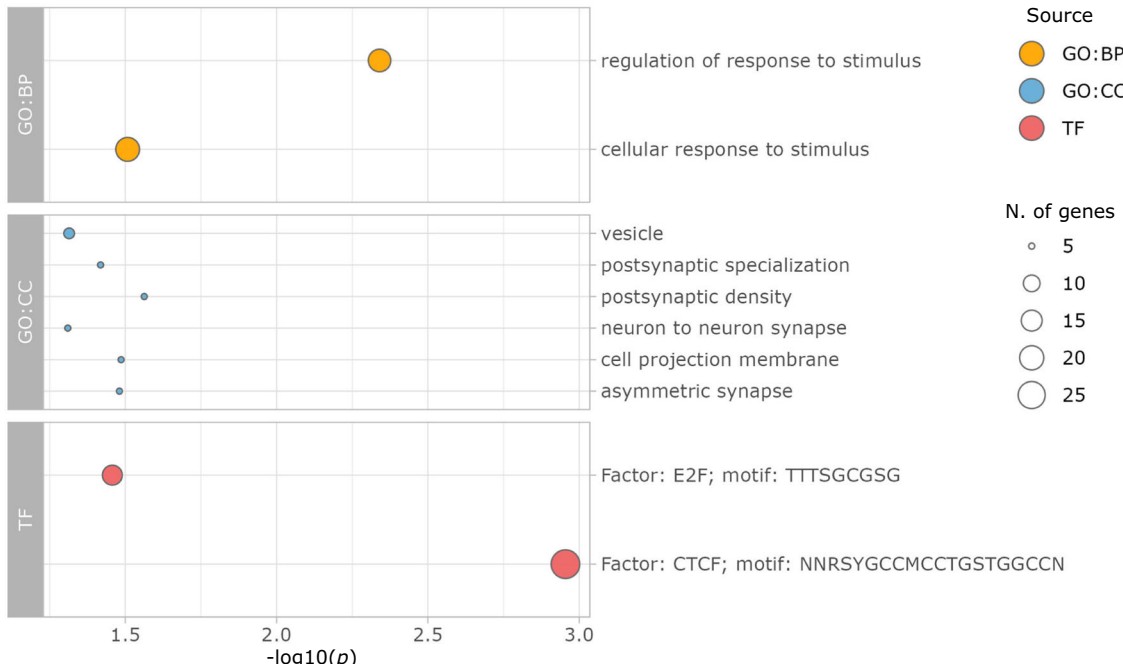

**Fig. 4 Pathway enrichment analysis in pathologically confirmed DLB.** Functional enrichment of significant gene ontology pathways for biological processes (GO:BP, orange dot), cellular components (GO:CC, blue dots), and TRANSFAC database (TF, red dots) in pathologically confirmed DLB versus controls. The x-axis shows the $p$ value associated with each pathway on a -$\log_{10}$ scale and the size of each dot indicates the number of genes involved.

| Table 3 Biological pathways associated with differential methylation in DLB. | | | | |
|---|---|---|---|---|
| **ID** | **Pathway** | **Source** | **No. Genes** | ***P* value** |
| GO:0048583 | regulation of response to stimulus | GO:BP | 18 | 4.57E-03 |
| GO:0051716 | cellular response to stimulus | GO:BP | 20 | 3.11E-02 |
| GO:0014069 | postsynaptic density | GO:CC | 5 | 2.74E-02 |
| GO:0031253 | cell projection membrane | GO:CC | 5 | 3.27E-02 |
| GO:0032279 | asymmetric synapse | GO:CC | 5 | 3.14E-02 |
| GO:0099572 | postsynaptic specialization | GO:CC | 5 | 3.82E-02 |
| GO:0031982 | vesicle | GO:CC | 6 | 4.85E-02 |
| GO:0098984 | neuron to neuron synapse | GO:CC | 5 | 4.90E-02 |
| TF:M12593 | Factor: CTCF; motif: NNRSYGCCMCCTGSTGGCCN | TF | 29 | 1.11E-03 |
| TF:M00918 | Factor: E2F; motif: TTTSGCGSG | TF | 14 | 3.49E-02 |

*No. Genes* number of genes, *GO* Gene Ontology, *BP* Biological Process, *CC* Cellular Component, *TF* TRANSFACT.
*P* value refers to Bonferroni corrected *p* value.

expressed in the vast majority of neurons, though Purkinje cells represent an exception. Our approach has already been successfully applied in EWAS based on DNA from cerebellar tissue[35,36]. Moreover, exploring the surrogate variables, we identified a mild negative correlation between NeuN-negative cell proportion and the surrogate variables 1 and 3 (Pearson correlation = −0.64 and −0.61, respectively), and a positive correlation between NeuN-positive cell proportion and surrogate variable 3 (Pearson correlation = 0.83). These data suggest that the surrogate variable analysis accounts for at least some of the variability due to the different cell types within a tissue. As such, it represents a valuable approach that could be employed in similar instances where there is no tool to estimate the cell proportion in a tissue (Supplementary Fig. 3).

Genome-wide association studies are affected by inflation and EWAS are not an exception, so exploring inflation is crucial to reduce the number of unreliable results. Our data showed moderate inflation (lambda = 1.19) (Fig. 1b). We further reduced genomic inflation using the R package *bacon* (lambda = 1.02, Supplementary Fig. 1), and we were able to replicate the results of

the main EWAS. To consolidate our main results, we also performed the EWAS using the MLM-based omic association (MOA) tool from OSCA[37]. This tool represents a stringent approach to processing DNA methylation data. Not surprisingly, therefore, inflation was drastically reduced when this tool was applied (lambda = 1.005), but none of the probes surpassed the genome-wide significance threshold (Supplementary Fig. 4). However, the top 48 CpG sites identified in our study showed a consistent pattern when comparing the two approaches (Pearson correlation = 0.87789), and the directions of their modulations were coherent (Supplementary Fig. 5).

Sample size differences between the male- and the female-specific cohorts may represent a bias of the study, particularly for the sex-specific EWAS. Furthermore, the absence of matching for age and sex between the patients and controls might have led to an overestimation of the contribution of the identified loci in the pathogenesis of DLB. However, this effect was probably mitigated by including age and sex as covariates in the association model[38]. The use of post-mortem tissues cannot discriminate between causal effects and the downstream

consequences of the DNA methylation changes observed. Finally, we detected only modest effects on DNA methylation, though this is consistent with previous EWAS efforts investigating neurodegenerative disorders[8,10,30].

In summary, we investigated the differential methylation signature of a large cohort of patients with pathologically confirmed DLB. We delineated clear epigenetic modulation associated with this common form of neurodegeneration, defined differentially methylated probes and regions, and highlighted new loci and biological pathways affected by these changes. In particular, we provide evidence implicating the *ARSB* gene and the CTCF-mediated synaptic plasticity to DLB. Our study underlines the potential role that epigenetic modulation plays in the pathogenesis of DLB and represents an opportunity for the future identification of biomarkers and new therapeutic targets.

## Methods

**Study samples**. Frozen cerebellar tissue from 298 DLB patients and 203 neurologically healthy controls were obtained from brain donation programs. The demographic and clinical characteristics of the study participants are summarized in Supplementary Table 1. The DLB patients were diagnosed with pathologically definite disease (limbic or neocortical subtype) according to the McKeith consensus criteria[1]. Neurologically healthy controls were selected based on the absence of neurological disease in their clinical history and the absence of neurodegenerative disease on pathological examination. All participants were of European ancestry. Informed consent for post mortem brain tissue donation was obtained from all subjects or their surrogate decision makers according to the Declaration of Helsinki. Each brain donation program was approved by its own institutional ethics committee. These convenience control samples were obtained from the same brain banks as the DLB cases and were of European ancestry. The controls were not specifically matched for age or sex, however, age and sex distributions among cases and controls were comparable (Supplementary Fig. 6, Supplementary Table 1).

**SNP genotyping, quality control, and data processing**. Genomic DNA was extracted from cerebellar tissue samples using the Maxwell RSC Tissue kit following the manufacturer's instructions (Promega, Madison, WI, USA). Each DNA sample was processed on the Infinium Global Diversity Array + Neuro Booster chip (v.1.0, Illumina, San Diego, CA, USA). This array genotypes over 1.8 million single-nucleotide polymorphisms (SNPs) across the genome, including 75,000 genetic variants previously implicated in common neurodegenerative diseases. Quality control checks were performed using PLINK (v.1.9.0-beta 4.4)[39]. Samples were excluded from the analysis based on the following criteria: (1) genome-wide call rate < 0.98, (2) heterozygosity outliers (> ± 0.15 F-statistic), (3) mismatch between reported sex and genotypic sex, (4) non-European ancestry (based on principal components analysis when compared to the HapMap 3 Genome Reference Panel), and (5) duplicates and related samples (pi-hat > 0.125). For variant-level quality control, we excluded variants for the following reasons: (1) non-random missingness between cases and controls ($p$ value $\leq 1\times10^{-4}$), (2) haplotype-based non-random missingness ($p$ value $\leq 1\times10^{-6}$), (3) deviation from Hardy-Weinberg equilibrium test in controls (mid-$p$ value $\leq 1\times10^{-6}$), and (4) overall missingness rate of $\geq 5\%$.

**DNA methylation profiling, quality control, and data processing**. For each sample, 450 ng of genomic DNA was bisulfite-converted (Zymo Research, Irvine, CA, USA) and processed on Infinium HumanMethylation EPIC BeadChips (v.1.0, Illumina),

according to the manufacturer's protocol. This array measures DNA methylation signals of 863,904 CpGs across the genome. Raw intensity data were processed in R (v.4.0.5) using the MethylAid (v.1.24.0)[40], minfi (v.1.36.0)[41], wateRmelon (v.1.34.0)[42], and maxprobes (v.0.0.2, https://github.com/markgene/maxprobes) packages.

Quality checks and filtering were performed to exclude low-quality samples and probes (summarized in Fig. S7). Samples meeting the following criteria were excluded: (1) low overall quality based on sample-dependent and sample-independent control probes and methylated/unmethylated probes ratio (using default settings in MethylAid), (2) bisulfite conversion rate < 80.0%, (3) mismatch between reported sex and genotypic sex, (4) mean detection $p$ value > 0.01, (5) samples that were flagged as outliers using the default setting in wateRmelon, and (6) samples that had > 1.0% of probes with a detection $p$ value > 0.05.

We then performed a principal component analysis of the Infinium MethylationEPIC array control probes. Data normalization was done using the R package minfi (preprocessFunnorm function), with 24 principal components (explaining 99.0% of variance)[43]. Following normalization, we excluded probes that had: (1) detection $p$ value < 0.01, (2) CpG sites containing SNPs of any minor allele frequency, (3) probes located on sex chromosomes, and (4) any Illumina methylation array 450 K and EPIC850K cross-reactive probes[44].

We generated beta- and M-values using the minfi functions getBeta and getM, respectively. Significant surrogate variables have been generated from M-values through the sva package (v.3.46.0) in R[45]. We estimated the proportion of neuronal and non-neuronal cell types using the minfi package (estimateCellCounts function)[35,36].

**EWAS analysis in DLB**. The DNA methylation status of each CpG site (as measured by $\beta$-values, Fig. S8) was tested for association with DLB using the Bioconductor package limma (v.3.46.0)[42] in R (v.4.0.5). In contrast to GWAS, there are no common guidelines established in the field for conducting EWAS analyses. We performed our study using beta values since they directly represent the proportion of methylated CpG sites, making it easier to relate the values to biological processes. Supporting this approach, beta values and M-values showed highly correlated outputs in our analysis (Pearson correlation of $p$ values = 0.904), and the top 48 probes showed a consistent direction of effect between the two approaches (Fig. S9). Age, sex, experimental batch, the first five principal components (generated from the Infinium Global Diversity Array + Neuro Booster genotyping data to account for population stratification), NeuN-positive/NeuN-negative cell type proportion (minfi), and all significant surrogate variables (n = 44) (Bioconductor package sva, v.4.3) were included as covariates in the linear regression model. A two-sided $p$ value and a $\Delta\beta$-value was calculated for each CpG site. We used the Illumina EPIC annotation R package (IlluminaHumanMethylationEPICanno.ilm10b4.hg19) to define the overlapping genes. Four additional genes were manually annotated since the probes overlapped a gene (*IQSEC1*: cg06951630, *LINC01158*:cg07171538, and *LINC00856*:cg05757757) or mapped within 10 base-pairs upstream of a gene (*EPHA8*:cg25394625). $P$ values were adjusted by Bonferroni correction. The Bonferroni threshold for declaring a differentially methylated probe (DMP) to be genome-wide significant was $6.87 \times 10^{-8}$ ( = 0.05/728,197 markers). We identified sub-significant probes surpassing the False Discovery Rate threshold of 0.05, and we included them in the functional enrichment analysis. To further reduce genomic inflation removing unknown bias, we corrected the $p$ values using the Bioconductor package bacon (v.1.26.0). EWAS evaluations were

also performed comparing male ($n = 165$ cases and 122 controls) and female participants ($n = 113$ cases and 50 controls) through an interaction model, to assess possible sex-specific epigenetic modulation. Sex was not included as a covariate in those analyses.

The Bioconductor package DMRcate (v.2.4.1) was used with the recommended default settings (lambda = 1000 and C = 2, corresponding to 1 standard deviation of Gaussian kernel each 500 base pairs) to identify and evaluate regions in the DLB data for evidence of differential methylation[43]. This software calculated $p$ values based on smoothed FDR of CpGs within the region.

**Pathway enrichment analysis.** Functional enrichment analysis was performed using the g:Profiler toolkit (v.0.7.0)[46]. We investigated the following pathways: 1) Gene Ontology (GO) biological processes ($n = 15,808$), 2) GO cellular components ($n = 1973$), 3) GO molecular functions ($n = 5015$)[47,48], 4) pathways from the Kyoto Encyclopedia of Genes and Genomes (KEGG; $n = 563$)[49–51], 5) pathways described in Reactome ($n = 2532$)[52], 6) WikiPathways ($n = 790$)[53], 7) transcription factors in the TRANSFAC database ($n = 11,647$)[54], 8) microRNAs in miRTarBase ($n = 2658$)[55], 9) proteins in the Human Protein Atlas ($n = 830$)[56], 10) protein complexes in CORUM ($n = 2885$)[57], and 11) traits in the Human Phenotype Ontology database ($n = 10,668$)[58]. Pathways with fewer than five genes and a $p$ value > 0.05 were excluded from the analysis. Bonferroni corrections were applied to the $p$ values in each pathway to correct for multiple testing.

**Statistics and reproducibility.** We performed a case-control association study by fitting a linear regression model for each marker using the Bioconductor package limma (v.3.46.0). The topTable function was used to calculate the statistics of differentially methylated probes comparing DLB cases to healthy control subjects, adjusting the $p$ values for multiple testing. Bonferroni-corrected genome-wide significance threshold was set to $p < 6.87 \times 10^{-8}$ ( = 0.05/728,197 sites tested). We applied a False Discovery Rate $p$ value correction to declare sub-significant markers. To facilitate reproducible results, we made the analysis code publicly available on Github (https://github.com/pireho/EWAS-Lewy_body_dementia) and https://zenodo.org (DOI: 10.5281/zenodo.10365334).

**Reporting summary.** Further information on research design is available in the Nature Portfolio Reporting Summary linked to this article.

## Data availability

The EWAS summary statistics have been deposited into the EWAS catalog (www.ewascatalog.org). Individual-level methylation array data are available on dbGaP (accession #: phs001963). Source data underlying Figs. 2 and 4 are provided in Supplementary Data 4.

## Code availability

Analyses were performed using open-source tools and the code is available on GitHub at https://github.com/pireho/EWAS-Lewy_body_dementia and https://zenodo.org (https://doi.org/10.5281/zenodo.10365334)[59].

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

## Acknowledgements

We thank the patients and families whose help and participation made this work possible. We are grateful to the Banner Sun Health Research Institute Brain and Body Donation Program of Sun City, Arizona, for the provision human biological materials. The Brain and Body Donation Program is supported by the National Institute of Neurological Disorders and Stroke (U24 NS072026 National Brain and Tissue Resource for Parkinson's Disease and Related Disorders), the National Institute on Aging (P30 AG19610 and P30 AG072980, Arizona Alzheimer's Disease Center), the Arizona Department of Health Services (contract 211002, Arizona Alzheimer's Research Center), the Arizona Biomedical Research Commission (contracts 4001, 0011, 05-901, and 1001 to the Arizona Parkinson's Disease Consortium) and the Michael J. Fox Foundation for Parkinson's Research. The study used tissue samples and data from the Johns Hopkins Morris K. Udall Center of Excellence for Parkinson's Disease Research (NIH P50 NS38377). The Parkinson's UK Brain Bank at Imperial College London is funded by Parkinson's UK, a charity registered in England and Wales (948776) and in Scotland (SC037554). We thank the members of the Laboratory of Neurogenetics (NIH) for their collegial support and technical assistance. We thank the NABEC Consortium, the Virginia Commonwealth University Brain Bank, and the NIH NeuroBioBank for providing tissue samples. Tissue/data used in this research was obtained from the Human Brain Collection Core, Intramural Research Program, NIMH (http://www.nimh.nih.gov/hbcc). This work utilized the computational resources of the NIH HPC Biowulf cluster (http://hpc.nih.gov). This study was supported by the Intramural Research Program of the National Institutes of Health (National Institute of Neurological Disorders and Stroke; project number: ZIANS003154). K.K. was funded by the Finnish Cultural Foundation, The Finnish Parkinson Foundation, The Päivikki and Sakari Sohlberg Foundation, and The Finnish Brain Foundation. This study was supported by the Intramural Research Program of the National Institutes of Health (National Institute of Neurological Disorders and Stroke; project number: ZIANS003154). K.K. was funded by the Finnish Cultural Foundation, The Finnish Parkinson Foundation, The Päivikki and Sakari Sohlberg Foundation, and The Finnish Brain Foundation.

## Author contributions

Conceptualization: P.R., S.S.-A., B.J.T., H.P.E., S.W.S.; Formal analysis: P.R., S.S.-A., P.Ru., Z.H., R.C., K.K.; Investigation: P.R., S.S.-A., P.Ru., S.S., Z.S., R.C., K.K.; Resources: B.S.T., S.M.G, A.K.H., D.A., E.S.M., K.L.N., R.W., M.S.A., T.M.D., L.S.R., J.C.T., O.P., G.E.S., T.G.B.; Data curation: P.R., S.S.-A., P.Ru., S.S., Z.S., R.C., K.K.; Writing – original draft: P.R.; Writing – review & editing: All authors; Visualization: P.R., S.S.-A., P.Ru.; Supervision, project administration and funding: S.W.S.

## Funding

## Competing interests

The authors declare the following competing interests: S.W.S. serves on the Scientific Advisory Council of the Lewy Body Dementia Association and the Multiple System Atrophy Coalition. S.W.S. and B.J.T. receive research support from Cerevel Therapeutics. S.W.S. serves on the editorial board of the journal JAMA Neurology. B.J.T. is associate editor for the journal Brain. B.J.T. holds patents on the clinical testing and therapeutic implication of the C9orf72 repeat expansion. All other authors have no conflicts of interest to declare that are relevant to the content of this article.
