## [Peer review file · Communications Biology]

Reviewers' comments:

Reviewer #1 (Remarks to the Author):

I thank the editor for the opportunity to review this manuscript by Reho et al. They report a methylome-wide association study of bulk cerebellar tissue in 278 brain bank donors with dementia with Lewy bodies (DLB) and 172 control donors. They present more than a thousand differentially methylated probes and an unspecified number of differentially methylated regions. Additional analyses include sex-specific association testing, investigation of candidate loci and pathway enrichment analysis.

The study addresses an important question and has a decent sample size relative to other publications in the field. The data generated are therefore of clear value. In my opinion however, the analysis has important shortcomings that would need to be addressed before publication could be considered.

Major:

The study investigates methylation in bulk cerebellar tissue. This means that the methylation profiles represent a mix of all cell types, and differences in cell composition is likely to be an important driver of any association signal unless the linear model is adjusted for cell composition. Cell composition could be estimated from methylation data and used as a covariate in the model, an approach that I'd currently consider standard in neurodegenerative brain EWAS.

The number of differentially methylated probes reported is very high, which makes me suspect that p-values in the analyses are inflated overall. This is a major issue in EWAS of complex disorders and needs to be addressed. The manuscript does not report a lambda value as a measure of inflation, nor is any Q-Q plot shown. P-value inflation could result from methodological bias or from large, general differences in the global methylation pattern comparing patients to controls. Current practice varies across studies with respect to how high lambda values are handled, but most larger studies aim to bring lambda down to some reasonable value, e.g. by estimating surrogate variables that are included in the linear model. An alternative approach, which controls for the global methylation pattern, is the OSCA tool for omic-data-based complex trait analysis, which is a quite conservative method that effectively brings lambda down towards 1. I would consider reporting lambda value, discussing these aspects of analysis and preferentially reporting how different methodological choices affect the results a necessary requirement for a rigorous EWAS analysis.

The data availability statement seems inadequate: The Communications Biology instruction for authors state: "An inherent principle of publication is that others should be able to replicate and build upon the authors' published claims. Please note that all published manuscripts reporting original research in Nature Portfolio journals must include a data availability statement. The data availability statement must make the conditions of access to the "minimum dataset" that are necessary to interpret, verify and extend the research in the article, transparent to readers." The expected current standard would be to make the full raw data available in an established repository, such as GEO. This would make the dataset truly valuable to the research community. If sharing of raw data through a public repository is for some reason impossible, this needs to be explained and justified, potentially outlining other ways that the data could be accessed by other researchers.

Furthermore, the authors seem to have misunderstood how the word "code" is used in the context of medical research, stating that "The code for the analysis is available at the associated website of each software package." This is far too general to be of any value. "Code" should be taken to mean the actual commands given by the analyst to process the specific data used in your study to produce the results presented in the manuscript.

Matching between cases and controls is not commented on, but from Supplementary Table 7, it is apparent that for males in particular, controls were on average as much as 10 years younger. Furthermore, controls could be as young as 38 years, whereas the youngest DLB patient was 57. Despite controlling for age in the regression model, this lack of age matching is a concerning

limitation that must be discussed.

A number of minor concerns could also be raised, but to limit the length of this review, I will not include all of these – only a few examples:

- The abstract and Results sections need to specify that the study has investigated tissue from the cerebellum
- I have not been able to find Supplementary Table 2? The text does not state the total number of DMRs
- For pathway enrichment analysis, it is stated that “Bonferroni corrections were applied to the p-values in each pathway to correct for multiple testing”, yet it doesn’t mention how many pathways were tested
- It is confusing that hypermethylation is in the negative direction in the volcano plots, yet in the positive direction in the violin plot.
- With respect to previous studies, the Discussion states: “Our study failed to replicate these previously published findings”, but it is unclear how this was assessed.
- The Discussion lacks reflections on the causal status of methylation changes
- The statement “prima facie evidence of clinical benefit if the disease-associated expression changes can be reversed” seems exaggerated.
- The abstract should use as N the number of donors that actually went into the analyses.

Reviewer #2 (Remarks to the Author):

This is an interesting and timely study exploring DNA methylation differences in the cerebellum in DLB cases compared to controls. The premise of the study is important, but I do have some concerns about the data analysis. The authors report 1,011 Bonferroni significant differentially methylated sites in DLB cases. This seems very high compared to other EWAS and the authors do comment on the fact that the various [other] studies may be underpowered at present (even though some are larger than the current one). They also add that their sample size was modest in size, hampering their ability to identify relevant loci, and so focus on genes previously implicated in DLB. However, this statement was not in line with their results which state they identified 1,011 Bonferroni significant findings. One contributor to the fact they found more loci than other similar/more powered studies is that the other studies have controlled for neuron/glia proportions which will be altered in disease. I acknowledge this is not straightforward in the cerebellum as many of these algorithms are based on neuN sorted neurons, which is a protein not expressed in pyramidal cells, but they should at least have discussed this limitation in the limitations section of their discussion.

My main concern with the study is that the number of Bonferroni significant loci seems far too high for an EWAS of this sample size based on previous brain EWAS studies and particularly as prior studies have shown in the case of Alzheimer’s disease (AD) that there are no DNA methylation alterations in the cerebellum (but are in other brain regions). This is not really discussed in the discussion although the previous studies are mentioned. It is important that the authors explore inflation in their model, and provide QQ plots and lambda to show that the model is not inflated. It would also be important to explore whether an interaction model may be a better way to look at the sex specific effects, to formally investigate whether there are male-driven alterations in disease.

It would be beneficial for the authors to provide a heatmap showing the relationship of the PCs (from the normalised methylation data) with all potential covariates to show how they ended up using these covariates.

A few other specific comments are provided below:

Line 92 – Need to state here that they used cerebellar brain tissue.

Line 99 – The authors say that the majority of DMPs map to genes that were located within the

gene body. Can they formally test for an enrichment of this by comparing to the proportion of total probes on the array?

Methods – DMR identification. It would be useful to state what the recommended default settings are.

Methods – Gene-set enrichment analyses. Need to define how they mapped the probes to the genes – is it based on Illumina annotation, or on a set distance from the gene

Methods – code availability – a link to the github page could be provided for reproducibility. In addition, a direct URL to the EWAS summary statistics would be beneficial here rather than the catalog.

Figure 2 is misleading – it needs to include the proportion of all probes passing QC, rather than total number as there are different coverage in different regions. What about probes in intergenic regions?

Figure 3 – It would be useful to explain what the parts of the violin plot explain (ie the bar, central line, and perimeters of the violin). It is unclear what is being illustrated in figures B, C, D. Figure D is also not mentioned in the legend

Figure 1 and Figure 4 – It would be useful to also include Manhattan plots (as supplementary figures)

Table 1 – The legend is difficult to follow and would benefit from further clarification, for example what does the $\Delta\beta$ value refer to, is this difference between DLB and control (ie 0.015 is 15%) and is negative representing hypomethylation in DLB?

Did the authors investigate whether DMPs were driven by genetic differences between groups?

Results – line 102-111. How did the authors define them as highly expressed in the brain? Was this by exploring on a public database?

GENERAL COMMENTS

We thank the editorial staff and the reviewers for their insightful and constructive comments on our manuscript. Below, we respond to the editorial and reviewer's remarks, which we believe have strengthened our article. Changes to the revised manuscript are highlighted in yellow.

EDITORIAL COMMENTS

1) Data availability statement required. The major asset for this paper is the dataset which has to be made available.

Response: We agree with the reviewers and the editors that the availability of the data is crucial for reproducible research. We deposited the summary statistics of our study into the EWAS catalog (www.ewascatalog.org). The individual-level methylation data generated in this study will be shared on dbGaP upon publication. We added the following data availability statement to our manuscript (page 12, paragraph 3, lines 341 – 342)

“The EWAS summary statistics have been deposited into the EWAS catalog (www.ewascatalog.org). Individual-level methylation array data are available on dbGaP (accession #: phs001963).”

2) Analysis code provision to ensure reproducibility.

Response: We have made the analysis code publicly available on GitHub, and we updated our manuscript to include the link (page 12, paragraph 3, lines 342 - 344):

“Analyses were performed using open-source tools and the code is available on GitHub at https://github.com/pireho/EWAS-Lewy_body_dementia.”

3) Detailed description of statistical analysis especially regarding inflation. Both reviewers are concerned about the large number of statistically significant results and the lack of QC to defend these results.

Response: We thank the reviewers for their constructive comments. We reprocessed our data and updated quality control metrics for our revised manuscript as follows: 1) we generated surrogate variables (using the *sva* package in R) and adjusted our data by using all 44 significant variables, 2) we estimated the cell type proportions (using *minfi* package in R), adjusting the data by NeuN-positive and NeuN-negative cell type, and 3) we performed a more stringent *p*-value correction through the R package *bacon*.

We updated the Methods section accordingly (page 10, paragraph 5, lines 289 - 292):

“We generated beta- and M-values using the *minfi* functions *getBeta* and *getM* respectively. Significant surrogate variables have been generated from M-values through the *sva* package (v.3.46.0) in R. We estimated the proportion of neuronal and non-neuronal cell types using the *minfi* package (*estimateCellCounts* function.^{35, 36}”

We updated the Methods section with the revised covariate information (page 11, paragraph 1, lines 297 - 300):

“Age, sex, experimental batch, the first five principal components (generated from the Infinium Global Diversity Array + Neuro Booster genotyping data to account for population stratification), NeuN-positive/NeuN-negative cell type (*minfi*), and all significant surrogate variables (n = 44) (Bioconductor package *sva*, v.4.3) were included as covariates in the linear regression model.”

Additional corrections were applied using the package *bacon*. We updated the Methods section as follows (page 11, paragraph 1, lines 309 - 310):

“To further reduce genomic inflation removing unknown bias, we corrected the *p*-values using the Bioconductor package *bacon* (v.1.26.0).”

We added a QQ-plot to the manuscript as Fig. 1b and indicate moderate residual inflation with a lambda of 1.19 as shown below:

To our knowledge, there are no tool sets for estimating the cell proportion using the cerebellum as reference. Instead, we used the *estimateCellCount* function of the *minfi* package. Although this function is not set for cerebellar tissue, we performed the analysis using dorsolateral prefrontal cortex as a reference, as previously described by Guintivano and colleagues (Epigenetics, 2013. PMID: 23426267), and Pellegrini et al. (Front Aging Neuroscience, 2021. PMID: 33790779). This tool discriminates the proportion of cells expressing the neuronal specific protein NeuN (NeuN_positive, NeuN_negative). Of note, Purkinje cells do not express this protein, representing a limitation of the approach that we added to the Discussion section (page 8, paragraph 4, lines 214 - 218):

“The *minfi*-approach that we used to estimate the cell type proportion was able to discriminate between neuronal and non-neuronal cells, using the frontal cortex region as reference. This method is based on the neuronal-specific protein NeuN, expressed in the vast majority of neurons, though Purkinje cells represent an exception. Our approach has already been successfully applied in EWAS based on DNA derived from cerebellar tissue.^{35, 36} However, it represents a limitation of our study.”

In summary, our approach allowed us to significantly reduce inflation. The lambda-value in the revised EWAS was 1.19, which is similar to other published EWAS studies (see for example an EWAS in type 2 diabetes: PMID: 35169870; EWAS in Alzheimer’s disease: PMID: 35982059; EWAS in asthma: 35033200). Analyzing these data, we identified seven differentially methylated probes (DMPs) in DLB patients. We updated Fig. 1 and Table 1 accordingly to reflect these updated analyses (see below).

Moreover, we also corrected the p -values using the package *bacon*, a particularly stringent approach to reduce genomic inflation. Through this method, we were able to further reduce inflation ($\lambda = 1.02$; Supplementary Fig. 4), identifying 2 out of 7 DMPs surpassing the Bonferroni significance threshold, and the 5 out of 7 DMPs surpassing the false discovery rate significance threshold. These findings showed that our first approach was strongly reliable, so we decided to keep it for our EWAS, considering $\lambda = 1.19$ an acceptable inflation level. This inflation level is consistent with other published EWAS studies (see for example an EWAS in type 2 diabetes: PMID 35169870; EWAS in Alzheimer's disease: PMID 35982059; EWAS in asthma: PMID 35033200). The Discussion has been updated accordingly to discuss this approach (page 8, paragraph 5, lines 219 - 224):

“Genome-wide association studies are affected by inflation and EWAS are not an exception, so exploring inflation is crucial to reduce the number of unreliable results. Our data showed moderate inflation ($\lambda = 1.19$) (Fig. 1b). We further reduced genomic inflation using the R package *bacon* ($\lambda = 1.02$, Supplementary Fig. 4), and we were able to replicate the results of the main EWAS. Sample size differences between the male- and the female-specific cohorts may represent a bias of the study, particularly for the sex-specific EWAS.”

We added Supplementary Fig. 4 to the Supplementary Information, which shows the results using *bacon*:

Supplementary Fig. 4. Volcano plot, QQ-plot, and Manhattan plot of the EWAS using *bacon*

Volcano plot (a) showing statistical significance ($-\log_{10} p$ -value) and magnitude of change ($\Delta\beta$) of all CpG sites included in the EWAS. Red dots indicate significantly hypomethylated DMPs, and hypermethylated CpGs are displayed as blue dots. The Bonferroni adjusted p -value < 0.05 threshold is shown as a red dashed line. QQ-plot (b) showing the p -value distribution and inflation (lambda values, top left). Density plot (c) illustrating the observed p -values distribution. Manhattan plot (d) demonstrating the p -values of the probes across the genome. The Bonferroni adjusted p -value < 0.05 threshold is shown as a red dashed line, while the orange dashed line represents the FDR-significance threshold. Probes surpassing the genome-wide significance are shown as red dots. P -values have been corrected using the Bioconductor package *bacon*.

4) a thorough discussion of lack of matching of cases and controls

Response: We have added the following sentence to the Methods describing how control samples were selected (page 9, paragraph 2, lines 248-251):

“These convenience control samples were obtained from the same brain banks as the DLB cases and were of European ancestry. The controls were not specifically matched for age or sex, however, age and sex distributions among cases and controls were comparable (Supplementary Fig. 1, Supplementary Table 1).”

We have added the following sentence to the Discussion describing the lack of matching based on age and sex as a potential limitation of our study (page 8, paragraph 5, lines 224 - 227). We also added a relevant reference on the matching in genetic studies:

“Furthermore, the absence of matching for age and sex between the patients and controls might have led to an overestimation of the contribution of the identified loci in the pathogenesis of DLB. However, this effect was probably mitigated by including age and sex as covariates in the association model.³⁷”

We have added the following plot comparing the age and sex distribution of the cases and controls as Supplementary Fig. 1:

“**Supplementary Fig. 1.** Age and sex distribution of DLB cases and controls used in the EWAS analysis.

Scatter plot comparing the age and sex distribution of the DLB cases and controls in our study cohort.

We have added the percentage of male and female cases and controls to column 4 of Supplementary Table 1:

Diagnosis	Sex	Mean Age at Death (Range)	N	Total
Control	M	67.5 yrs (38-97)	134 (66.0%)	203
	F	74.5 yrs (50-99)	69 (34.0%)	
DLB	M	78 yrs (57-99)	177 (59.4%)	298
	F	78 yrs (60-99)	121 (40.6%)	

5) Both reviewers also have several suggestions for improved analyses which should also be completed.

Response: We thank the editor for their comments. Please see our response to the reviewers' comments below on how we improved the analyses of our study.

REVIEWER 1

1. I thank the editor for the opportunity to review this manuscript by Reho et al. They report a methylome-wide association study of bulk cerebellar tissue in 278 brain bank donors with dementia with Lewy bodies (DLB) and 172 control donors. They present more than a thousand differentially methylated probes and an unspecified number of differentially methylated regions. Additional analyses include sex-specific association testing, investigation of candidate loci and pathway enrichment analysis.

The study addresses an important question and has a decent sample size relative to other publications in the field. The data generated are therefore of clear value.

Response: We thank the reviewer for their kind summary of our study.

2. In my opinion however, the analysis has important shortcomings that would need to be addressed before publication could be considered. The study investigates methylation in bulk cerebellar tissue. This means that the methylation profiles represent a mix of all cell types, and differences in cell composition is likely to be an important driver of any association signal unless the linear model is adjusted for cell composition. Cell composition could be estimated from methylation data and used as a covariate in the model, an approach that I'd currently consider standard in neurodegenerative brain EWAS.

Response: We thank the reviewer for raising this insightful point. We reanalyzed our data estimating the cell type count using the R package *minfi* (*estimateCellCount* function). Although this function is not set

for cerebellar tissue, we performed the analysis using frontal cortex as a reference (DLPFC), as previously described by Guintivano and colleagues (Epigenetics, 2013. PMID: 23426267), and Pellegrini et al. (Front Aging Neuroscience, 2021. PMID: 33790779). Using cell type and surrogate variables as covariates, we were able to reduce inflation and improve our analysis.

Please see our response to the editor's comments, point 3, above for additional details.

3. The number of differentially methylated probes reported is very high, which makes me suspect that p-values in the analyses are inflated overall. This is a major issue in EWAS of complex disorders and needs to be addressed. The manuscript does not report a lambda value as a measure of inflation, nor is any Q-Q plot shown. P-value inflation could result from methodological bias or from large, general differences in the global methylation pattern comparing patients to controls. Current practice varies across studies with respect to how high lambda values are handled, but most larger studies aim to bring lambda down to some reasonable value, e.g. by estimating surrogate variables that are included in the linear model. An alternative approach, which controls for the global methylation pattern, is the OSCA tool for omic-data-based complex trait analysis, which is a quite conservative method that effectively brings lambda down towards 1. I would consider reporting lambda value, discussing these aspects of analysis and preferentially reporting how different methodological choices affect the results a necessary requirement for a rigorous EWAS analysis.

Response: We thank the reviewer for raising this point. We re-analyzed our data, adjusting by cell-type and surrogate variables. We included QQ-plots and lambda values in the manuscript (see Fig. 1b for the overall analysis, Fig. 3b for the sex-specific EWAS, and Supplementary Fig. 4b for EWAS results using *bacon* as correction). Please also see our response to the editor's comments, points 2 and 3, above for additional details.

4. The data availability statement seems inadequate: The Communications Biology instruction for authors state: "An inherent principle of publication is that others should be able to replicate and build upon the authors' published claims. Please note that all published manuscripts reporting original research in Nature Portfolio journals must include a data availability statement. The data availability statement must make the conditions of access to the "minimum dataset" that are necessary to interpret, verify and extend the research in the article, transparent to readers." The expected current standard would be to make the full raw data available in an established repository, such as GEO. This would make the dataset truly valuable to the research community. If sharing of raw data through a public repository is for some reason

impossible, this needs to be explained and justified, potentially outlining other ways that the data could be accessed by other researchers.

Response: We thank the reviewer for raising this important point. We deposited the summary statistics from our EWAS into the EWAS catalog. The individual-level methylation data generated in this study will be shared on dbGaP upon publication. We updated the data sharing section as follows (page 12, paragraph 3, lines 341-342):

“The EWAS summary statistics have been deposited into the EWAS catalog (www.ewascatalog.org). Individual-level methylation array data are available on dbGaP (accession #: phs001963).”

5. Furthermore, the authors seem to have misunderstood how the word “code” is used in the context of medical research, stating that “The code for the analysis is available at the associated website of each software package.” This is far too general to be of any value. “Code” should be taken to mean the actual commands given by the analyst to process the specific data used in your study to produce the results presented in the manuscript.

Response: We thank the reviewer for raising this important point. The code has been published on Github and a link has been added to the manuscript (page 12, paragraph 3, lines 342 - 344):

“Analyses were performed using open-source tools and the code is available on GitHub at https://github.com/pireho/EWAS-Lewy_body_dementia.”

6. Matching between cases and controls is not commented on, but from Supplementary Table 7, it is apparent that for males in particular, controls were on average as much as 10 years younger. Furthermore, controls could be as young as 38 years, whereas the youngest DLB patient was 57. Despite controlling for age in the regression model, this lack of age matching is a concerning limitation that must be discussed.

Response: We agree with the reviewer that this is an important point and have added this limitation to the Discussion. Please see our response to the editor’s comments, point 4, above for additional details.

7. A number of minor concerns could also be raised, but to limit the length of this review, I will not include all of these – only a few examples: The abstract and Results sections need to specify that the study has investigated tissue from the cerebellum.

Response: Thank you for raising this point. We altered the Abstract to emphasize that the study was performed using cerebellar tissue (page 3, lines 45 - 47):

“We performed genome-wide DNA methylation mapping of cerebellar tissue obtained from 298 pathologically confirmed DLB cases and 203 controls to study the epigenetic profile of this understudied disease.”

We modified the Introduction to emphasize that we used cerebellar tissue as the basis of our study (page 4, paragraph 3, lines 80 - 83):

“We performed an epigenome-wide association study (EWAS) to characterize the differential methylation patterns in cerebellar tissue obtained from 298 pathologically confirmed DLB cases and 203 neurologically healthy controls.”

Similarly, we updated the beginning of the Results section (page 4, lines 88 - 89):

“We performed an EWAS using cerebellar brain tissue obtained from patients diagnosed with DLB and healthy individuals.”

8. I have not been able to find Supplementary Table 2? The text does not state the total number of DMRs

Response: We have ensured that all the supplementary tables were included in the revised submission.

We have also added the total number of DMRs to the Results as follows (page 6, paragraph 2, lines 134-136):

“Our analysis identified 32 CpG sites that clustered in three different DMRs characterized by a hypermethylation signature in DLB cases compared to controls (Table 2).”

9. For pathway enrichment analysis, it is stated that “Bonferroni corrections were applied to the p-values in each pathway to correct for multiple testing”, yet it doesn't mention how many pathways were tested.

Response: Thank you for pointing this out. In response to this comment, we expanded the description of the investigated pathways in our Methods section (page 11, paragraph 3, lines 320 - 327):

“We investigated the following pathways: 1) Gene Ontology (GO) biological processes (n = 15,808), 2) GO cellular components (n = 1,973), 3) GO molecular functions (n = 5,015),^{46, 47} 4) pathways from the Kyoto Encyclopedia of Genes and Genomes (KEGG; n = 563),^{48, 49, 50} 5) pathways described in Reactome (n = 2,532),⁵¹ 6) WikiPathways (n = 790),⁵² 7) transcription

factors in the TRANSFAC database (n = 11,647),⁵³ 8) microRNAs in miRTarBase (n = 2,658),⁵⁴ 9) proteins in the Human Protein Atlas (n = 830),⁵⁵ 10) protein complexes in CORUM (n = 2,885),⁵⁶ and 11) traits in the Human Phenotype Ontology database (n = 10,668).⁵⁷

10. It is confusing that hypermethylation is in the negative direction in the volcano plots, yet in the positive direction in the violin plot.

Response: We agree with the reviewer's point and updated the volcano plot so that the hypermethylated probes are in the positive direction (blue dots) and the hypomethylated probes are in the negative direction (red dots) (see for example Fig. 1a below):

11. With respect to previous studies, the Discussion states: “Our study failed to replicate these previously published findings”, but it is unclear how this was assessed.

Response: We agree with the reviewer that it is important to show these results. For this reason, we amended the Discussion (page 8, paragraph 2, line 201):

“Our study failed to replicate these previously published findings (Supplementary Table 4).”

We added Supplementary Table 4 in which we list the summary statistics from our study at previously described sites:

Supplementary Table 4. Differentially methylated probes and regions previously associated with DLB.

Previously reported probes/regions					Summary Statistics (this study)				
Probe_ID	Chr.	Position (hg19)	Overlapping genes	Source	Delta beta	CI.L	CI.R	P.Value	adj.P.Val
cg01920334	chr21	48,087,978	-	Pihlstrøm L, et al. (PMID: 35995800)	0.002197626	-0.0059755	0.01037079	0.59735599	1
cg06653480	chr22	45,034,765	-	Pihlstrøm L, et al. (PMID: 35995800)	-0.004970944	-0.0127985	0.0028566	0.21257412	1
cg19772267	chr14	63,754,790	RHOJ	Pihlstrøm L, et al. (PMID: 35995800)	0.000999652	-0.0032611	0.00526042	0.6448602	1
cg07837822	chr13	109,740,415	MYO16	Pihlstrøm L, et al. (PMID: 35995800)	-0.004108146	-0.0102387	0.00202246	0.18845563	1
cg07107199	chr1	205,215,911	TMCC2	Pihlstrøm L, et al. (PMID: 35995800)	-0.001917851	-0.0061335	0.00279784	0.37164715	1
cg14511218	chr10	6,962,843	-	Pihlstrøm L, et al. (PMID: 35995800)	-0.003533956	-0.0167379	0.00966994	0.59904445	1
cg02673002	chr11	74,459,317	RNF169	Pihlstrøm L, et al. (PMID: 35995800)	-0.000575444	-0.0047937	0.00364284	0.78868468	1
cg04147843	chr11	32,085,613	-	Pihlstrøm L, et al. (PMID: 35995800)	0.001811262	-0.0051871	0.00880963	0.6111522	1
cg15586054	chr6	166,072,030	PDE10A	Pihlstrøm L, et al. (PMID: 35995800)	0.000610066	-0.0012449	0.002465	0.5182617	1
cg16821230	chr3	53,647,443	CACNA1D	Pihlstrøm L, et al. (PMID: 35995800)	1.21534E-05	-0.0015314	0.00155567	0.98765689	1
cg09985192	chr14	32,797,255	AKAP6	Pihlstrøm L, et al. (PMID: 35995800)	0.006203164	-0.0046763	0.01708265	0.2629819	1
cg11700456	chr12	1,849,881	ADIPOR2	Pihlstrøm L, et al. (PMID: 35995800)	-0.00686381	-0.0128145	-0.0009131	0.02389076	1
cg19793404	chr3	87,842,899	-	Pihlstrøm L, et al. (PMID: 35995800)	-0.000948894	-0.0100071	0.00810928	0.83693316	1
cg22516775	chr16	79,071,140	WWOX	Pihlstrøm L, et al. (PMID: 35995800)	-0.001736778	-0.0040792	0.00660576	0.14574381	1

12. The Discussion lacks reflections on the causal status of methylation changes.

Response: We thank the reviewer for this comment. Our EWAS aimed to identify differential methylation probes and regions in a case-control study using association tests. It is premature to speculate about causation based on association results alone. Additional functional studies would be needed to test the causal relationships of the observed associations, which would go beyond the scope of the current study.

13. The statement “prima facie evidence of clinical benefit if the disease-associated expression changes can be reversed” seems exaggerated.

Response: We have deleted this line from the revised version.

14. The abstract should use as N the number of donors that actually went into the analyses.

Response: We have modified the Abstract to read as follows (page 3, lines 47 - 49):

“After quality control filtering, 728,197 CpG sites in 278 cases and 172 controls were available for the analysis.”

REVIEWER 2

1. This is an interesting and timely study exploring DNA methylation differences in the cerebellum in DLB cases compared to controls. The premise of the study is important, but I do have some concerns about the data analysis. The authors report 1,011 Bonferroni significant differentially methylated sites in DLB cases. This seems very high compared to other EWAS and the authors do comment on the fact that

the various [other] studies may be underpowered at present (even though some are larger than the current one). They also add that their sample size was modest in size, hampering their ability to identify relevant loci, and so focus on genes previously implicated in DLB. However, this statement was not in line with their results which state they identified 1,011 Bonferroni significant findings. One contributor to the fact they found more loci than other similar/more powered studies is that the other studies have controlled for neuron/glia proportions which will be altered in disease. I acknowledge this is not straightforward in the cerebellum as many of these algorithms are based on neuN sorted neurons, which is a protein not expressed in pyramidal cells, but they should at least have discussed this limitation in the limitations section of their discussion.

Response: We thank the reviewer for raising this insightful point. We reanalyzed our data estimating the cell type count using the R package *minfi* (*estimateCellCount* function). Although this function is not set for cerebellar tissue, we performed the analysis using frontal cortex as a reference (DLPFC), as previously described by Guintivano and colleagues (Epigenetics, 2013. PMID: 23426267), and Pellegrini et al. (Front Aging Neuroscience, 2021. PMID: 33790779). Using cell type and surrogate variables as covariates, we were able to reduce inflation and improve our analysis.

Please see our response to the editor's comments, point 3, above for additional details.

2. My main concern with the study is that the number of Bonferroni significant loci seems far too high for an EWAS of this sample size based on previous brain EWAS studies and particularly as prior studies have shown in the case of Alzheimer's disease (AD) that there are no DNA methylation alterations in the cerebellum (but are in other brain regions). This is not really discussed in the discussion although the previous studies are mentioned. It is important that the authors explore inflation in their model, and provide QQ plots and lambda to show that the model is not inflated. It would also be important to explore whether an interaction model may be a better way to look at the sex specific effects, to formally investigate whether there are male-driven alterations in disease.

Response: We thank the reviewer for raising this insightful point. We re-analyzed the data, adjusting by cell-type and surrogate variables. We included QQ-plots and lambda values in the manuscript (Fig. 1, Fig. 3, Supplementary Fig. 4). Please see our response to the editor's comments, point 3, and Reviewer 1, point 3.

We re-processed the sex-specific EWAS through an interaction model, and we edited the Results section accordingly (page 6, paragraph 3, lines 142 - 144):

“To explore differential methylation patterns among men and women, we performed an **interaction model comparing** male study participants ($n = 165$ DLB cases and $n = 122$ controls) and female subjects ($n = 113$ DLB cases and $n = 50$ controls).”

We also updated the Materials and Methods section (page 11, paragraph 1, lines 310 - 313):

“EWAS evaluations were also performed **comparing** male ($n = 165$ cases and 122 controls) and female participants ($n = 113$ cases and 50 controls) **through an interaction model**, to assess possible sex-specific epigenetic modulation. Sex was not included as a covariate in those analyses.”

3. It would be beneficial for the authors to provide a heatmap showing the relationship of the PCs (from the normalised methylation data) with all potential covariates to show how they ended up using these covariates.

Response: We thank the reviewer for this suggestion. We tested the interaction between all available covariates with 10 principal components derived from DNA methylation data. We added the requested figure as Supplementary Fig. 5:

4. Line 92 – Need to state here that they used cerebellar brain tissue.

Response: We now clearly state that we used cerebellar brain tissue in the Abstract, Introduction, and the Results sections, as well as the Methods.

Please also see our response to Reviewer 1, point 7, above.

5. Line 99 – The authors say that the majority of DMPs map to genes that were located within the gene body. Can they formally test for an enrichment of this by comparing to the proportion of total probes on the array?

Response: The statement no longer applies to the re-analyzed data set, as the gene body enrichment is not detected in seven differentially methylated probes.

6. Methods – DMR identification. It would be useful to state what the recommended default settings are.

Response: We thank the reviewer for the comment. We expanded the methods section as follows (page 11, paragraph 2, lines 314 - 316):

“ The Bioconductor package *DMRcate* (v.2.4.1) was used with the recommended default settings ($\lambda=1000$ and $C=2$, corresponding to 1 standard deviation of Gaussian kernel each 500 base pairs) to identify and evaluate regions in the DLB data for evidence of differential methylation.”

7. Methods – Gene-set enrichment analyses. Need to define how they mapped the probes to the genes – is it based on Illumina annotation, or on a set distance from the gene.

Response: The statement does not apply to the re-analyzed data set, as we only have 7 DMPs that were significant after applying more stringent corrections. A gene-set enrichment analysis was not feasible on this small number of probes.

8. Methods – code availability – a link to the github page could be provided for reproducibility. In addition, a direct URL to the EWAS summary statistics would be beneficial here rather than the catalog.

Response: We thank the reviewer for this comment. The analysis code is available at:
https://github.com/pireho/EWAS-Lewy_body_dementia

The data from the revised analysis were submitted to the EWAS catalog, but we do not yet have an accession code. We hope that the accession code will be available soon. Below, we provide screenshots from our successful submission to the EWAS catalog:

Metadata submission

Summary statistics submission:

Please also see our responses to Reviewer 1, points 4 and 5.

9. Figure 2 is misleading – it needs to include the proportion of all probes passing QC, rather than total number as there are different coverage in different regions. What about probes in intergenic regions?

Response: The statement does not apply to the re-analyzed data set, and we removed this figure from the revised manuscript. We thank the reviewer for pointing this out.

10. Figure 3 – It would be useful to explain what the parts of the violin plot explain (ie the bar, central line, and perimeters of the violin). It is unclear what is being illustrated in figures B, C, D. Figure D is also not mentioned in the legend.

Response: In the revised version of the manuscript, we show the violin plot in Fig. 2 now. The legend has been expanded as follows (page 13, paragraph 2, lines 357 - 366):

“Fig. 2. Differentially methylated probes in DLB.

The violin plots show the DNA methylation (beta value) distribution (violin shape) in the seven differentially methylated probes. The vertical axis represents the range of values in the dataset, where 0 and 1 mean fully unmethylated and fully methylated respectively. The box plot represents the interquartile range of the dataset (25% bottom, 75% top), the middle line represents the median of the distribution, and the central line shows the value distribution. Black dots represent outliers. The overall difference between DLB cases and controls is shown as delta beta ($\Delta\beta$); negative and positive values refer to hypomethylation and hypermethylation in DLB cases, respectively (e.g., -0.020 indicates that DLB cases show a 2% decrease in DNA methylation compared to controls). *P*-values refer to Bonferroni corrected *p*-values.”

11. Figure 1 and Figure 4 – It would be useful to also include Manhattan plots (as supplementary figures).

Response: We thank the reviewer for this suggestion. We now include the Manhattan plots along with the volcano and QQ plots in the manuscript (please see Fig. 1d, Fig. 3d, and Supplementary Fig. 4d). As an example, we show the Manhattan plot in Fig. 1d here:

12. Table 1 – The legend is difficult to follow and would benefit from further clarification, for example

what does the $\Delta\beta$ value refer to, is this difference between DLB and control (ie 0.015 is 15%) and is negative representing hypomethylation in DLB?

Response: We thank the reviewer for raising this point. We expanded the legend in Table 1 as follows:

“Differentially methylated probes in the DLB EWAS. Chromosome positions are shown relative to the human reference genome (hg19). Gene names are shown according to UCSC RefGen. The $\Delta\beta$ values refer to the difference between DNA methylation (β -values) in cases compared to control (e.g., -0.020 indicates that DLB cases show a 2% decrease in DNA methylation compared to controls). Adjusted p -values refers to Bonferroni corrected p -values. Abbreviation: Chr., chromosome; $\Delta\beta$, delta beta; Adj.P, adjusted p -value.”

13. Did the authors investigate whether DMPs were driven by genetic differences between groups? Results – line 102-111. How did the authors define them as highly expressed in the brain? Was this by exploring on a public database?

Response: We thank the reviewer for these questions. We investigated methylation QTLs to identify a possible driving effect of genetics on DNA methylation in our study cohort. However, none of the DMPs reaching the genome-wide significance or sub-significance (False Discovery Rate adjusted p -value) thresholds showed a differential methylation driven by genetic differences. For this reason, we decided to do not include this analysis in the manuscript.

For the second question, we examined the expression profile of genes that harbored differentially methylated probes in brain tissues published within the GTEx portal (<https://www.gtexportal.org/home/>). This investigation demonstrated notable expression changes across various brain regions. We updated the Discussion (page 7, paragraph 2, lines 171 - 172) and added Supplementary Fig. 6 to illustrate this point:

“Interestingly, many of these genes are highly expressed in the brain (Supplementary Fig. 6) ...”

OTHER CHANGES

- 1) We removed Supplementary Tables 5-8 from the revised manuscript, which are no longer applicable.
- 2) We updated the authorship order to account for the contributions of new analyses in the revised manuscript. As such, Dr. Sara Saez-Atienzar has been moved up to the second position in the authorship order.
- 3) Following the journal formatting guidelines, we added a paragraph titled ‘statistics and reproducibility’ to the Methods section (page 12, first paragraph, lines 331 - 338):

“Statistics and reproducibility

We performed a case-control association study by fitting a linear regression model for each marker using the Bioconductor package *limma* (v.3.46.0). The *topTable* function was

used to calculate the statistics of differentially methylated probes comparing DLB cases to healthy control subjects, adjusting the p -values for multiple testing. Bonferroni-corrected genome-wide significance threshold was set to $p < 6.87 \times 10^{-8}$ ($= 0.05/728,197$ sites tested). We applied a False Discovery Rate p -value correction to declare sub-significant markers. To facilitate reproducible results, we made the analysis code publicly available on Github (https://github.com/pireho/EWAS-Lewy_body_dementia).”

Reviewers' comments:

Reviewer #1 (Remarks to the Author):

Thanks to the editor for the opportunity to re-review this DLB bulk cerebellum EWAS manuscript by Reho et al.

The most important update from the previous version of this manuscript is that the authors have done appropriate changes to the analysis workflow following the first round of reviews. Remarkably, this has brought the number of significant CpGs reported down from more than a thousand to seven (!), highlighting the importance of rigorous methodology. Reflecting this profound difference in the main results, the revised manuscript is largely a completely new paper.

I find that the manuscript is substantially improved. There are however, still some issues to address before I would recommend publication.

The authors have estimated cell type proportions and surrogate variables, which has successfully brought the genomic inflation factor (λ) down to an acceptable level, demonstrating a reasonable QQ-plot. Still, as these are crucial steps in the analysis pipeline and essential for the interpretation of the results, I believe more detail and consideration is warranted here:

- The authors state in the Methods that "significant surrogate variables have been generated from M-values", yet the differential methylation analysis itself was based on beta-values. What justifies this discrepancy?

- 44 SVs is a very high number. Surrogate variables are assumed to capture factors such as unrecorded batch effects, which would not plausibly be as many as 44. It would be valuable to explore if top SVs correlate with e.g. cell type proportions, which might indicate that some of these SVs are capturing additional variability in cell type composition not accounted for by the simple dichotomous estimation of neuronal vs non-neuronal cells.

- As mentioned in my first review, OSCA is a tool that effectively controls inflation in EWAS studies, representing an alternative to surrogate variable analysis. As the estimated number of SVs is so high, I believe it would strengthen the paper to also repeat the association analysis with OSCA to see whether this method yields results consistent with the standard limma approach.

- I think it is important to show whether estimated cell type proportions were associated with disease status or not. Furthermore, cell type proportions have not been included in the correlation plot of Supplementary figure 5, where it would be a variable of particular interest.

In my first review, I requested more detail on the comparison with previous studies. The authors have now included a supplementary table, but I still find the Discussion very limited and superficial when it comes to putting the results into context of other published work. This is particularly important as the presented study has no independent replication. The specific findings would therefore be regarded as tentative, and comparison with other studies, discussion of reproducibility etc, is in my view more important than a detailed discussion of tentative top-hits, so if word count is a limitation, this could be downscaled in my opinion.

There is no mention of specifically how the different published studies differ from this one with respect to sample size, brain region, design etc. For instance, Shao et al probably has the most equivalent design to this study, but assessed methylation in Brodmann area 7 and included only 31 patients. Nasamran et al. compared PDD and DLB in blood whereas Pihlstrom et al. studied frontal cortex with Braak Lewy body stage as outcome. These differences are important, and stating that the study "failed to replicate" these findings when designs are that different is in my view directly misleading. This should be prioritized more in the discussion. Perhaps a plot could be used to indicate whether there is a general tendency towards similar directions of effect with Shao et al, which is the most similarly designed EWAS.

The authors also reference an Alzheimer's disease meta-GWAS in this part of the Discussion

(Smith et al.), yet it does not seem that they actually try to compare the results with this one. Given the known overlap between DLB and AD, and the much larger sample sizes achieved to date for AD EWAS, this would also be interesting.

In my previous review, I also suggested the authors reflect on causality of methylation changes, where they reply that "it is premature to speculate about causation based on association results alone". This is of course true, and my intention was to highlight this as an important limitation of any postmortem EWAS. The authors claim that their study "highlight the critical contributions of methylation to the pathogenesis of DLB" and "underlines the vital role that epigenetic modulation plays in the pathogenesis" - but the methylation differences observed may for all we know be downstream consequences of end-stage disease and not "critical contributions" at all. I maintain that this caveat needs to be mentioned as a major limitation - although of course general to any postmortem EWAS.

I also find the Discussion imbalanced with respect to the choice of cerebellum as the region to study. The authors write that "selecting a relatively spared tissue source provides a more accurate window into the epigenetic plasticity of a disease". This might be true, but needs to be weighed against the fact that the disease-relevant changes are likely more prominent and representative in the regions primarily affected by the disease - so data on cerebellum alone is also a significant limitation. Sampling multiple regions for comparison would of course have been ideal.

Minor final point: The authors should take care to write "cell type proportions" and not merely "cell type" as on page 4, line 93. This could confuse the reader into thinking that the input was not bulk tissue but different cell types.

Reviewer #3 (Remarks to the Author):

The authors thoroughly address the methodological issues raised by both reviewers particularly the question of bias and statistical inflation. By adjusting the regression with age, sex, experimental batch, five principal components from genotyping data, cell type, and 44 surrogate variables from methylation data. This has reduced the number of significant sites to 7. The authors addressed all other technical and presentation issues raised by both reviewers adequately.

Two minor points:

Line 101 the values should be corrected from "-0.100 to 0.091" should change to "-0.01 to 0.091"
Line 153. It is unclear what does the statement "the analysis did not identify DMR associated with sex-specific epigenetic modulations" mean?

GENERAL COMMENTS

We thank the reviewers for their constructive comments on our manuscript. Below, we provide a point-by-point response to the reviewers' comments. Any changes to the revised manuscript are highlighted in yellow.

RESPONSES TO REVIEWER 1

1. The most important update from the previous version of this manuscript is that the authors have done appropriate changes to the analysis workflow following the first round of reviews. Remarkably, this has brought the number of significant CpGs reported down from more than a thousand to seven (!), highlighting the importance of rigorous methodology. Reflecting this profound difference in the main results, the revised manuscript is largely a completely new paper. I find that the manuscript is substantially improved. There are, however, still some issues to address before I would recommend publication.

The authors have estimated cell type proportions and surrogate variables, which has successfully brought the genomic inflation factor (λ) down to an acceptable level, demonstrating a reasonable QQ-plot.

Response: We thank the reviewer for their kind summary of our revised study.

2. The authors state in the Methods that "significant surrogate variables have been generated from M-values", yet the differential methylation analysis itself was based on beta-values. What justifies this discrepancy?

Response: We thank the reviewer for raising this point. The Bioconductor package "sva" that we used for this analysis is designed to work with M-values or beta values. We used the M-values, the log-transformed ratios of methylated probe intensity to unmethylated probe intensity, to generate the surrogate variables. This preference is due to several reasons:

1. M-values have better statistical properties for differential analysis compared to beta values. They are more normally distributed and homoscedastic.

2. Studies have shown that using M-values increases the sensitivity and specificity in detecting differential methylation.
3. M-values proved a more linear relationship with the methylation proportion.
4. M-values help in stabilizing the variance across different levels of methylation.

While we preferred M-values for the initial statistical reasons, we used beta-values for the downstream analysis because of their more intuitive biological interpretation. Beta values range from 0 to 1 and are more intuitive as they directly represent the proportion of methylated CpG sites, making it easier to relate their values to biological processes. Indeed, in our study, the beta values and M-values were highly correlated (Pearson correlation of p -values = 0.904); furthermore, the M-values and beta values of the top 48 CpGs displayed a consistent correlation, as shown in the figure below. In light of this, we believe that it is reasonable to have generated the surrogate variables from the M-values, and then use beta values for all of the downstream analyses.

We edited the manuscript to emphasize these statistical points (page 11, lines 327-330; page 12, lines 331-333):

“In contrast to GWAS, there are no common guidelines established in the field for conducting EWAS analyses. We therefore performed our study using beta values since they directly represent the proportion of methylated CpG sites, making it easier to relate the values to biological processes. Supporting this approach, beta values and M-values showed highly correlated outputs in our analysis (Pearson correlation of p -values = 0.904), and the top 48 probes showed a consistent direction of effect between the two approaches (Fig. S4).”

We added Figure S4 to the Supplementary Materials:

Fig S4. Graph comparing the M-values and beta values of the top 48 CpG sites (i.e., probes that were FDR significant). The M-values and beta values were normalized using a Z-score computed as delta beta (or M) / standard deviation. The red dots denote probes that were significant after the Bonferroni correction.

3. 44 SVs is a very high number. Surrogate variables are assumed to capture factors such as unrecorded batch effects, which would not plausibly be as many as 44. It would be valuable to explore if top SVs correlate with e.g. cell type proportions, which might indicate that some of these SVs are capturing additional variability in cell type composition not accounted for by the simple dichotomous estimation of neuronal vs non-neuronal cells.

Response: We thank the reviewer for this suggestion. We did not randomly pick the number of surrogate variables to be included. Instead, the *sva* package in R computes the list of significant surrogate variables (SV) to be included in the analysis. This approach reduced inflation, as was requested by the reviewer in their first round of comments.

We also investigated a second approach in which the *sva* package returns the number of surrogate variables that you could use. Based on our study design, the tool computed seven SVs. However, adjusting our data for only seven SVs resulted in inflated data. Therefore, following their suggestion, we used the more stringent, conservative method for our EWAS analysis including all significant SVs.

In response to the reviewer's request, we investigated the correlation between SVs and other covariates. We found a negative correlation between NeuN-negative cell proportion and surrogate variables 1 and 3, and a positive correlation between NeuN-positive cell proportion and surrogate variable 3.

We edited the manuscript as follows (page 9, lines 237-243):

“Moreover, exploring the surrogate variables, we identified a mild negative correlation between NeuN-negative cell proportion and the surrogate variables 1 and 3 (Pearson correlation = -0.64 and -0.61, respectively), and a positive correlation between NeuN-positive cell proportion and surrogate variable 3 (Pearson correlation = 0.83). These data suggest that the surrogate variable analysis accounts for at least some of the variability due to the different cell types within a tissue. As such, it represents a valuable approach that could be employed in similar instances where there is no tool to estimate the cell proportion in a tissue (Figure S8).”

4. As mentioned in my first review, OSCA is a tool that effectively controls inflation in EWAS studies, representing an alternative to surrogate variable analysis. As the estimated number of SVs is so high, I believe it would strengthen the paper to also repeat the association analysis with OSCA to see whether this method yields results consistent with the standard limma approach.

Response: We thank the reviewer for this suggestion. We repeated the analysis using the OSCA-MOA tool (see reference <https://yanglab.westlake.edu.cn/software/osca/#MOA>). This approach represents a very stringent option for EWAS, meaning that none of the probes achieved genome-wide significance in the analysis, as shown below.

However, we observed consistent directions of effect when we compared the most significant variants from the OSCA and Limma approaches (Figure S5). The Pearson correlation of the p -values for the top 48 FDR-significant probes in both approaches was 0.878.

Figure S5. Comparison the limma and OSCA-MOA EWAS results

Comparison between top 48 CpG sites (FDR significant probes) obtained from OSCA-MOA and Limma approaches. Data points were transformed as log-fold change divided by the standard error. Red dots show the top 7 probes that surpassed the Bonferroni threshold for genome-wide significance.

We edited the manuscript and updated the code accordingly (page 9, lines 247-254):

“To consolidate our main results, we also performed the EWAS using the MLM-based omic association (MOA) tool from OSCA.³⁷ This tool represents a stringent approach to processing DNA methylation data. Not surprisingly, therefore, inflation was drastically reduced when this tool was applied ($\lambda = 1.005$), but none of the probes surpassed the genome-wide significance threshold (Figure S6). However, the top 48 CpG sites identified in our study showed a consistent pattern when comparing the two approaches (Pearson correlation = 0.87789), and the directions of their modulations were coherent (Figure S5).”

5. I think it is important to show whether estimated cell type proportions were associated with disease status or not. Furthermore, cell type proportions have not been included in the correlation plot of Supplementary figure 5, where it would be a variable of particular interest.

Response: We thank the reviewer for this point. We revised this figure, which now includes the cell type proportions in the correlation plot (see figure below). Our analysis showed no significant correlation between cell proportion and the disease status (PHENO) of the subjects (Pearson correlation coefficient NeuN_neg/phenotype = -0.02, and NeuN_pos/phenotype = -0.09).

Figure S8. Correlation plot of DNA methylation principal components and covariates

6. In my first review, I requested more detail on the comparison with previous studies. The authors have now included a supplementary table, but I still find the Discussion very limited and superficial when it comes to putting the results into context of other published work. This is particularly important as the presented study has no independent replication. The specific findings would therefore be regarded as tentative, and comparison with other studies, discussion of reproducibility etc., is in my view more important than a detailed discussion of tentative top-hits, so if word count is a limitation, this could be downscaled in my opinion.

Response: We thank the reviewer for the comment. Please see our replies to your specific points below.

7. There is no mention of specifically how the different published studies differ from this one with respect to sample size, brain region, design etc. For instance, Shao et al probably has the most equivalent design to this study, but assessed methylation in Brodmann area 7 and included only 31 patients. Nasamran et al. compared PDD and DLB in blood whereas Pihlstrom et al. studied frontal cortex with Braak Lewy body stage as outcome. These differences are important, and stating that the

study "failed to replicate" these findings when designs are that different is in my view directly misleading. This should be prioritized more in the discussion. Perhaps a plot could be used to indicate whether there is a general tendency towards similar directions of effect with Shao et al, which is the most similarly designed EWAS.

Response: We thank the reviewer for raising this point. We edited the manuscript as follows (page 8, lines 198-208):

~~“A recent EWAS in DLB showed a DNA methylation signature and highlighted differentially methylated probes and regions in the blood and frontal cortex. Our study failed to replicate these previously published findings (Supplementary Table 4). This disagreement may be because of differences in the tissue or brain region assayed in each of these studies. Only a few studies have investigated the epigenetic changes associated with DLB, and they differ in sample size, targeted tissue, and study design.^{8, 10, 30} For example, Shao and colleagues performed an EWAS of the Brodman area 7 of the brain in a cohort consisting of fifteen pathologically confirmed DLB cases and sixteen neurologically healthy controls. Despite the limited sample size and the different brain regions investigated, their study design does represent the closest structure to our EWAS. In contrast, Nasamran and collaborators profiled blood epigenetic modulations comparing 42 DLB patients and 50 Parkinson’s disease dementia cases, while Pihlstrom and colleagues explored the epigenetic modulations associated with different Braak Lewy body disease stages in 322 Parkinson’s disease and DLB cases. Our study identified none of the DMPs or DMRs previously associated with DLB (Table S3). However, the outlined differences in the design of these studies may account for these discrepancies.”~~

The authors also reference an Alzheimer's disease meta-GWAS in this part of the Discussion (Smith et al.), yet it does not seem that they actually try to compare the results with this one. Given the known overlap between DLB and AD, and the much larger sample sizes achieved to date for AD EWAS, this would also be interesting.

Response: We thank the reviewer for this comment. We agree that comparing Alzheimer's disease and DLB results is interesting. We modified Supplementary Data 3, adding the DMPs and DMRs previously identified by Smith and collaborators. We edited the manuscript accordingly (page 8, lines 209-219):

“Significant changes in methylation have also been observed in Alzheimer's disease and Parkinson's disease.^{31, 32, 33} Sharma and collaborators showed that epigenetic modifications across single nucleotide polymorphisms, located within the first intron of the *SNCA* gene, modulate the susceptibility to Parkinson's disease. A meta-analysis of 1,453 individuals with Alzheimer's disease, investigating the epigenetic changes associated with Braak neurofibrillary tangle stage, identified differentially methylated sites and regions in the prefrontal cortex, temporal gyrus, and entorhinal cortex, but not in the cerebellum. Interestingly, our EWAS replicated some of the findings from Smith's study, identifying epigenetic modification involving a common CpG site within *FRMD4A* (cg03775372) and a common DMP-associated gene, *MKL2*. This finding raises the scientific interest surrounding cg03775372 and the *FRMD4A* gene, since the same CpG reached the FDR significance in our main study and achieved genome-wide significance in the sex-specific EWAS.”

8. In my previous review, I also suggested the authors reflect on causality of methylation changes, where they reply that "it is premature to speculate about causation based on association results alone". This is of course true, and my intention was to highlight this as an important limitation of any postmortem EWAS. The authors claim that their study "highlight the critical contributions of methylation to the pathogenesis of DLB" and "underlines the vital role that epigenetic modulation plays in the pathogenesis" - but the methylation differences observed may for all we know be downstream consequences of end-stage disease and not "critical contributions" at all. I maintain that this caveat needs to be mentioned as a major limitation - although of course general to any postmortem EWAS.

Response: We agree with the reviewer on this point and made the following changes to our manuscript to elaborate on the limitations of studying postmortem tissue:

Discussion (page 7, line 170):

“...highlight the **potential** contributions of methylation to the pathogenesis of DLB...”

Discussion (page 9, lines 259-261):

“**The use of post-mortem tissues cannot discriminate between causal effects and the downstream consequences of the DNA methylation changes observed.**”

Discussion (page 10, line 268):

“...underlines the **potential** role that epigenetic modulation plays in the pathogenesis...”

9. I also find the Discussion imbalanced with respect to the choice of cerebellum as the region to study. The authors write that "selecting a relatively spared tissue source provides a more accurate window into the epigenetic plasticity of a disease". This might be true, but needs to be weighed against the fact that the disease-relevant changes are likely more prominent and representative in the regions primarily affected by the disease - so data on cerebellum alone is also a significant limitation. Sampling multiple regions for comparison would of course have been ideal.

Response: Thank you for this constructive comment. In the revised version of our manuscript, we expanded our Discussion as follows (page 8, lines 223-235):

“**This detail needs to be weighed against the fact that the disease-relevant changes are likely more prominent and representative in the regions primarily affected by the disease. Sampling multiple regions for comparison would have been ideal.**”

10. Minor final point: The authors should take care to write "cell type proportions" and not merely "cell type" as on page 4, line 93. This could confuse the reader into thinking that the input was not bulk tissue but different cell types.

Response: Thank you for this comment. We updated our manuscript accordingly to avoid any confusion.

RESPONSE TO REVIEWER 2

1. The authors thoroughly address the methodological issues raised by both reviewers particularly the question of bias and statistical inflation. By adjusting the regression with age, sex, experimental batch, five principal components from genotyping data, cell type, and 44 surrogate variables from methylation data. This has reduced the number of significant sites to 7. The authors addressed all other technical and presentation issues raised by both reviewers adequately.

Response: We thank the reviewer for their kind summary of our study.

2. Line 101 the values should be corrected from “-0.100 to 0.091” should change to “ -0.01 to 0.091”
Line 153. It is unclear what does the statement” the analysis did not identify DMR associated with sex-specific epigenetic modulations” mean?

Response: Thank you for these suggestions. Beta values ranged from -0.1 to + 0.091, so we cannot change the values as suggested.

We edited the manuscript to clarify the second point raised by the reviewer as follows (page 6, lines 153-154):

“Finally, our sex-specific EWAS did not identify any significant differentially methylated regions.”

OTHER CHANGES

1. We updated Figure S9 to include all 48 FDR-significant probe-associated genes:
Figure S9. Expression plot of the differentially methylated genes

Heatmap plot showing the expression pattern of the differentially methylated genes in different brain regions obtained from GTEx database (<https://www.gtexportal.org>). The light yellow boxes show low expression genes, while dark blue boxes identify highly expressed genes. Abbreviations: Transcripts Per Million, TPM.